# Durvalumab and cediranib with and without olaparib in recurrent ovarian cancer: a phase II proof-of-concept study

Junya Tabata [1,8], Tzu-Ting Huang [1,8] ✉, Elena Giudice [1,2], Kristen R. Ibanez[1,3], Jayakumar R. Nair[1], Aanika Balaji Warner [1], Britanny B. Solarz[1], Valentina Bolanos[1], Bernadette Redd [4], Nahoko Sato [5], Shraddha Rastogi [5], Sunmin Lee [5], Roshan L. Shrestha[5], Alexander Y. Mitrophanov [6], Stanley Lipkowitz [1], Kevin Conlon[1], Chien Chu Huang[1] & Jung-Min Lee[1,7] ✉

Here we report the efficacy and translational findings of durvalumab, olaparib, and cediranib (D + O + C) and of durvalumab plus cediranib (D + C) from the recurrent ovarian cancer cohort within a single-center, multi-arm, non-randomized, multi-cohort phase I/II trial (NCT02484404). Sixty-eight patients were enrolled (39 in D + O + C, 29 in D + C). The primary endpoint was objective response rate (ORR); secondary endpoints included progression-free survival (PFS) and safety. ORR was 19.4% (95% CI, 9.5-43.5) for D + O + C and 29.6% (95% CI, 13.8-46.9) for D + C; D + C met the primary endpoint while D + O + C did not. Median PFS was 4.5 months in both arms, with four exceptional responders (PFS ≥ 12 months) per arm. Toxicity was manageable. Pre- and on-treatment biopsies and blood samples were collected for prespecified transcriptomic and immunophenotypic profiling; signature analyses and preclinical studies were conducted post hoc and were exploratory. Baseline tumors from exceptional responders and patients with clinical benefit (partial response or stable disease with PFS ≥ 4 months) demonstrated enrichment of immune activation and metabolic pathways, whereas tumors with no clinical benefit (NCB; progressive disease or stable disease with PFS < 4 months) exhibited upregulation of vascular adaptation and cytoskeletal remodeling pathways. These findings support the proof-of-concept clinical activity of D + O + C and D + C and identify molecular signatures with potential predictive value in subsets of recurrent ovarian cancer.

Epithelial ovarian carcinoma (EOC) represents the most lethal gynecologic malignancy worldwide[1]. Nearly 75% of patients present with advanced disease, and more than two-thirds (~70%) experience disease recurrence within 3 years[2]. Standard of care (SOC) therapy for relapsed EOC involves platinum-based combination treatment (with paclitaxel, gemcitabine, or pegylated liposomal doxorubicin [PLD])[3,4], with or without bevacizumab for platinum-sensitive disease[5,6]. In the platinum-resistant setting, bevacizumab in combination with paclitaxel, PLD, or

[1]Women's Malignancies Branch, Center for Cancer Research, National Cancer Institute, Bethesda, MD, USA. [2]Division of Gynecologic Oncology, Humanitas San Pio X, Milan, Italy. [3]Department of Medicine, Icahn School of Medicine at Mount Sinai Morningside West Hospitals, New York, NY, USA. [4]Department of Radiology and Imaging Sciences, Clinical Center, National Cancer Institute, Bethesda, MD, USA. [5]Developmental Therapeutics branch, Center for Cancer Research, Clinical Center, National Cancer Institute, Bethesda, MD, USA. [6]Frederick National Laboratory for Cancer Research, National Institutes of Health, Frederick, MD, USA. [7]Present address: GlaxoSmithKline, Collegeville, PA, USA. [8]These authors contributed equally: Junya Tabata, Tzu-Ting Huang.
✉ e-mail: tzu-ting.huang@nih.gov; jamie9298@gmail.com

topotecan has demonstrated improved progression-free survival (PFS) but limited overall survival (OS) benefit[7]. Mirvetuximab soravtansine (MIRV), a folate receptor α (FRα)-targeting antibody-drug conjugate (ADC), was the first ADC to show an OS advantage compared to SOC in this setting, but its activity is restricted to tumors with high FRα expression and high-grade serous histology[8]. As such, additional therapeutic options are needed for the remaining platinum-resistant EOC population. Nab-paclitaxel plus relacorilant has demonstrated PFS improvement; however, this regimen was evaluated only in bevacizumab-pretreated platinum-resistant disease and excluded patients with primary platinum-refractory EOC[9]. These restrictions highlight the ongoing need for alternative therapeutic strategies in relapsed EOC.

Poly(ADP-ribose) polymerase inhibitors (PARPis) have been approved by the United States Food and Drug Administration (FDA) and the European Medicines Agency (EMA) as maintenance therapy for newly diagnosed EOC responding to platinum-based chemotherapy[10,11]. However, their use in relapsed settings has diminished[12], as regulatory agencies have restricted later-line use due to the lack of long-term survival benefit observed in the SOLO-3, ARIEL4, and QUADRA trials[13]. Although these studies demonstrated improvements in PFS, their inability to generate durable responses highlights the limitations of PARPi monotherapy in recurrent EOC and reinforces the need for rational combination strategies.

Immune checkpoint inhibitors (ICIs) targeting the programmed death-ligand 1 (PD-L1)/programmed cell death protein 1 (PD-1) axis have been evaluated in EOC, yet monotherapy has not demonstrated superiority over SOC treatments[14,15]. The limited activity reflects the profoundly immunosuppressive ovarian tumor microenvironment[16] and suggests the need for combination approaches. Recently, data from the ENGOT-ov65/KEYNOTE-B96 phase III study showed that adding pembrolizumab to weekly paclitaxel, with or without bevacizumab, significantly improved both PFS and OS in patients with platinum-resistant EOC[17], providing clinical evidence that ICIs can be effective when paired with an appropriate therapeutic backbone.

Preclinical studies indicate that PARPis may enhance tumor immunogenicity by increasing cytotoxic T-cell infiltration[18], activating the cyclic GMP-AMP synthase-stimulator of interferon genes (cGAS-STING) pathway[19], and promoting the accumulation of unrepaired DNA damage that recruits antigen-presenting cells[20]. Anti-angiogenic therapy further contributes to immune remodeling by reducing immunosuppressive cell populations within the EOC microenvironment[21]. Together, these mechanisms suggest that combining PARPi and anti-angiogenic therapy may create a tumor milieu more permissive to immune checkpoint blockade and thereby augment the antitumor activity of ICIs.

Early-phase clinical trials combining the vascular endothelial growth factor receptor 1–3 (VEGFR1-3) inhibitor cediranib with the PARPi olaparib demonstrated activity in a subset of patients with platinum-resistant EOC[22]. A subsequent phase I study incorporating durvalumab into olaparib and cediranib combination confirmed the tolerability of the triplet regimen and suggested preliminary clinical activity in recurrent EOC[23]. However, the multicenter phase II trial (NRG-GY023; NCT04739800) did not demonstrate superior activity of the triplet regimen compared with SOC (median PFS, 8.3 months versus 7.5 months)[24]. Notably, a subset of patients receiving ICI plus cediranib, with or without olaparib, achieved exceptional responses (PFS ≥ 12 months), including three patients (6.8%) in the durvalumab, olaparib, and cediranib (D + O + C) arm and two (4.8%) in the durvalumab plus cediranib (D + C) arm. However, this trial lacked translational analyses to characterize these responders[24].

In this study, we present clinical outcomes and translational findings from the recurrent EOC cohort within a single-center, multi-arm, non-randomized, multi-cohort phase I/II trial (NCT02484404). We performed transcriptomic profiling of pretreatment biopsies and

immunophenotypic analysis of serial blood samples, integrated with preclinical validation. Using parallel, treatment arm-specific analyses, we identified molecular signatures associated with treatment response (immune-primed and immunometabolic axes) versus resistance (alternative vascularization and cytoskeletal plasticity) in anti-VEGF/ICI ± PARPi regimens. These exploratory, hypothesis-generating findings suggest the potential of integrating immune, metabolic, and cytoskeletal profiling to guide biomarker-driven treatment strategies in recurrent EOC.

## Results

### Patient enrollment and baseline characteristics

Between September 2016 and August 2024, 68 recurrent ovarian cancer patients were enrolled and received at least one dose of treatment (D + O + C arm: 39 patients enrolled between November 2017 and August 2024; D + C arm: 29 patients enrolled between September 2016 and May 2023) (Fig. 1a, b). Baseline characteristics of each arm are detailed in Supplementary Data 1. Briefly, the D + O + C arm enrolled patients with EOC histology (high-grade serous [87.2%] and clear cell [12.8%]). The majority of patients (~85%) had BRCA wild-type and platinum-resistant or primary platinum-refractory disease. Also, a majority (~80%) had received prior bevacizumab. Previously, we reported the findings of the durvalumab plus olaparib (D + O) arm, which had similar baseline clinical characteristics, except for a lower rate of prior bevacizumab exposure (46%)[25].

The D + C arm also enrolled patients with mostly EOC histology (high-grade serous [72.4%], clear cell [6.9%], and high-grade endometrioid [3.4%]). 86.2% of patients were BRCA wild-type, and 69% of participants presented with platinum-resistant or primary platinum-refractory disease. About 59% of patients had received prior bevacizumab. The median number of prior treatments was three for both the D + O + C and D + C arms, representing a heavily pretreated population.

### Efficacy and safety

For the D + O + C arm, 7 of 36 (19.4%, 95% confidence interval [CI]: 9.5–43.5) Response Evaluation Criteria in Solid Tumors version 1.1 (RECIST v1.1)-evaluable patients achieved partial response (PR), including one unconfirmed PR, and 26 patients (72.2%) had stable disease (SD) as their best overall response (Fig. 1b). In the D + C arm, 8 of 27 (29.6%, 95% CI, 13.8–46.9) RECIST-evaluable patients achieved PR, including two unconfirmed PRs, and 13 (48.1%) had SD (Fig. 1b). Per the prespecified Simon two-stage design[26], the seven PRs observed in D + O + C did not meet the ≥11-response threshold, whereas the eight PRs observed in D + C exceeded the ≥6-response requirement, indicating promising activity despite under-enrollment.

The clinical benefit (CB, PR or SD ≥4 months) rate was 66.7 % (7 PRs and 17 SD ≥4 months; 24/36, 95% CI: 49.0–81.4) in the D + O + C arm, and 59.3% (8 PRs and 8 SD ≥4 months; 16/27, 95% CI, 38.8–77.6) in the D + C arm (Fig. 2a). Waterfall plots summarizing best tumor response in RECIST-evaluable patients are shown in Fig. 2a.

PRs were observed in both treatment arms irrespective of platinum sensitivity or prior bevacizumab exposure. Of the seven participants with PRs in the D + O + C arm, four had primary platinum-resistant disease, and three were platinum-sensitive. All seven PRs occurred in bevacizumab-pretreated patients. In the D + C arm, four were platinum-resistant (primary platinum-resistant [$n = 1$] and secondary platinum-resistant [$n = 3$]), one was primary platinum-refractory, and three were platinum-sensitive. Two PRs were observed in bevacizumab-pretreated patients.

In the intention-to-treat population, the median PFS was 4.5 months (CI: 3.8–6.1) for the D + O + C arm ($n = 39$), and 4.5 months (CI: 3.6–8.0) for the D + C arm ($n = 29$) (Fig. 2b). At the time of data cut-off (May 13, 2025), there were two patients with ongoing responses (17.9+ months [PR] in the D + O + C arm and 24.0+ months [PR] in the D + C

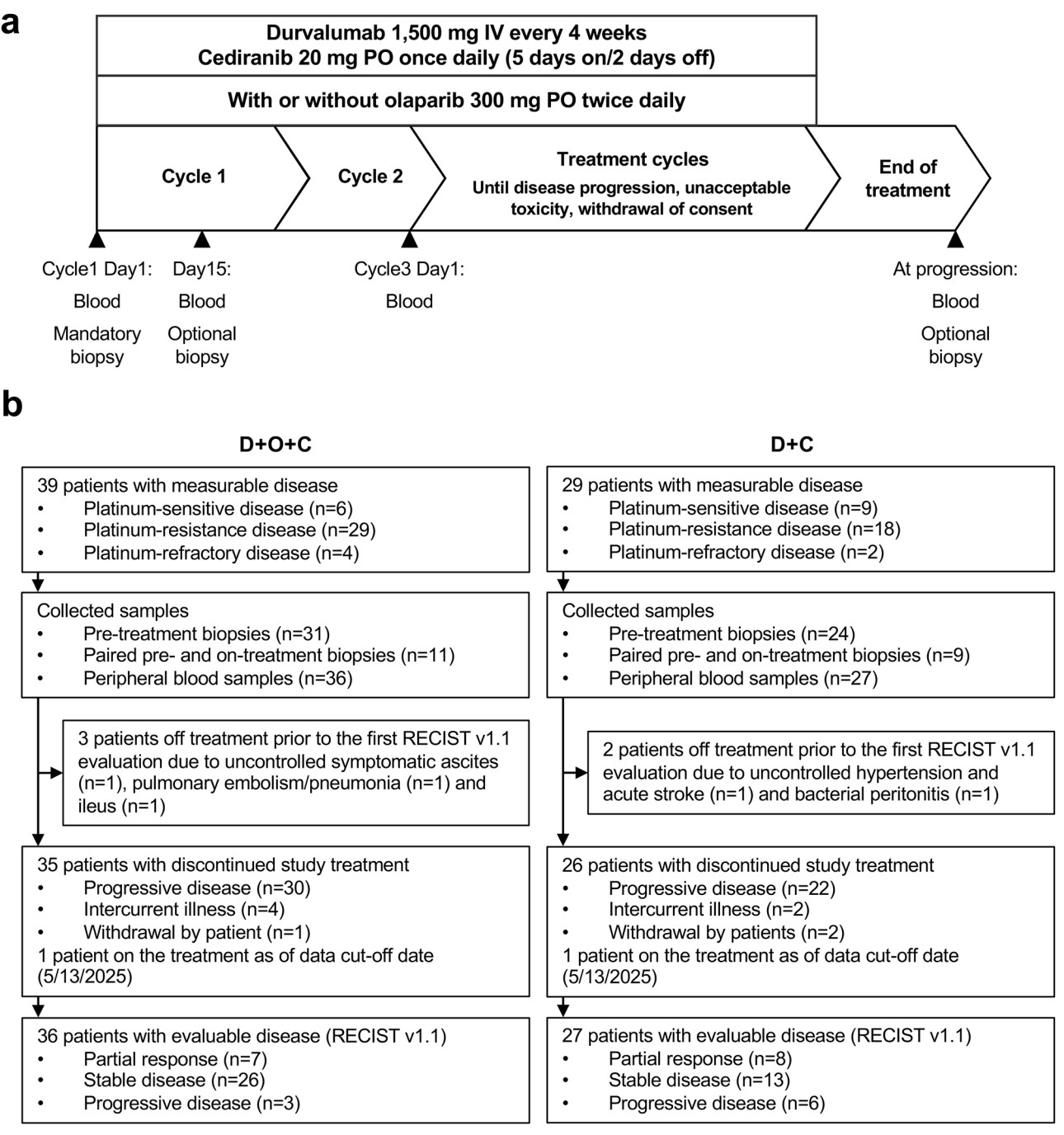

**Fig. 1 | Clinical trial design with integrated exploratory analysis. a** Overview of the clinical study schema and planned correlative analyses. Peripheral blood samples were collected prior to C1D1, C1D15, C3D1, and at disease progression. Tumor samples were collected prior to C1D1 (mandatory) and C1D15 (optional) and at progression (optional). CT scans were performed at baseline and every two cycles (±1 week) following treatment for RECIST evaluation. **b** CONSORT diagram summarizing patient enrollment and availability of clinical and correlative samples.

D + O + C arm: 39 enrolled, 36 RECIST-evaluable; 34 pretreatment biopsies and 11 on-treatment biopsies available. D + C arm: 29 enrolled, 27 RECIST-evaluable; 25 pretreatment biopsies and 9 on-treatment biopsies available. Reasons for early discontinuation before the first imaging are annotated. Source data are provided as a Source Data file. Abbreviations: C1D1, cycle 1 day 1; C1D15, cycle 1 day 15; C3D1, cycle 3 day 1; D + O + C, durvalumab, cediranib, and olaparib; D + C durvalumab plus cediranib, IV, intravenous, PO orally.

arm). Notably, there were four exceptional responders (defined as those with PFS ≥12 months) in the D + O + C arm (45.4 months [PR], 18.4 months [SD], 17.9 months [PR], and 13.6 months [SD]) and four in the D + C arm (24.0 months [PR], 23.5 months [PR], 13.2 months [SD], and 12.9 months [PR]). PFS for each patient is illustrated in Fig. 2b.

Treatment-related adverse events (TRAEs) occurring in ≥10% of patients are presented in Supplementary Data 3. Consistent with prior reports[24], the most common TRAEs observed in both treatment arms were hematologic and gastrointestinal. The most frequent all-grade

TRAEs were anemia (79.5%, 31/39) in the D + O + C arm and diarrhea (58.6%, 17/29) in the D + C arm.

### Upregulation of immune and metabolic pathways in patients with clinical benefit

To identify transcriptomic characteristics of patient tumors with CB, we first analyzed prespecified RNA sequencing (RNAseq) data from pretreatment fresh biopsy samples. Post hoc exploratory analyses using gene set enrichment analysis (GSEA) were performed to

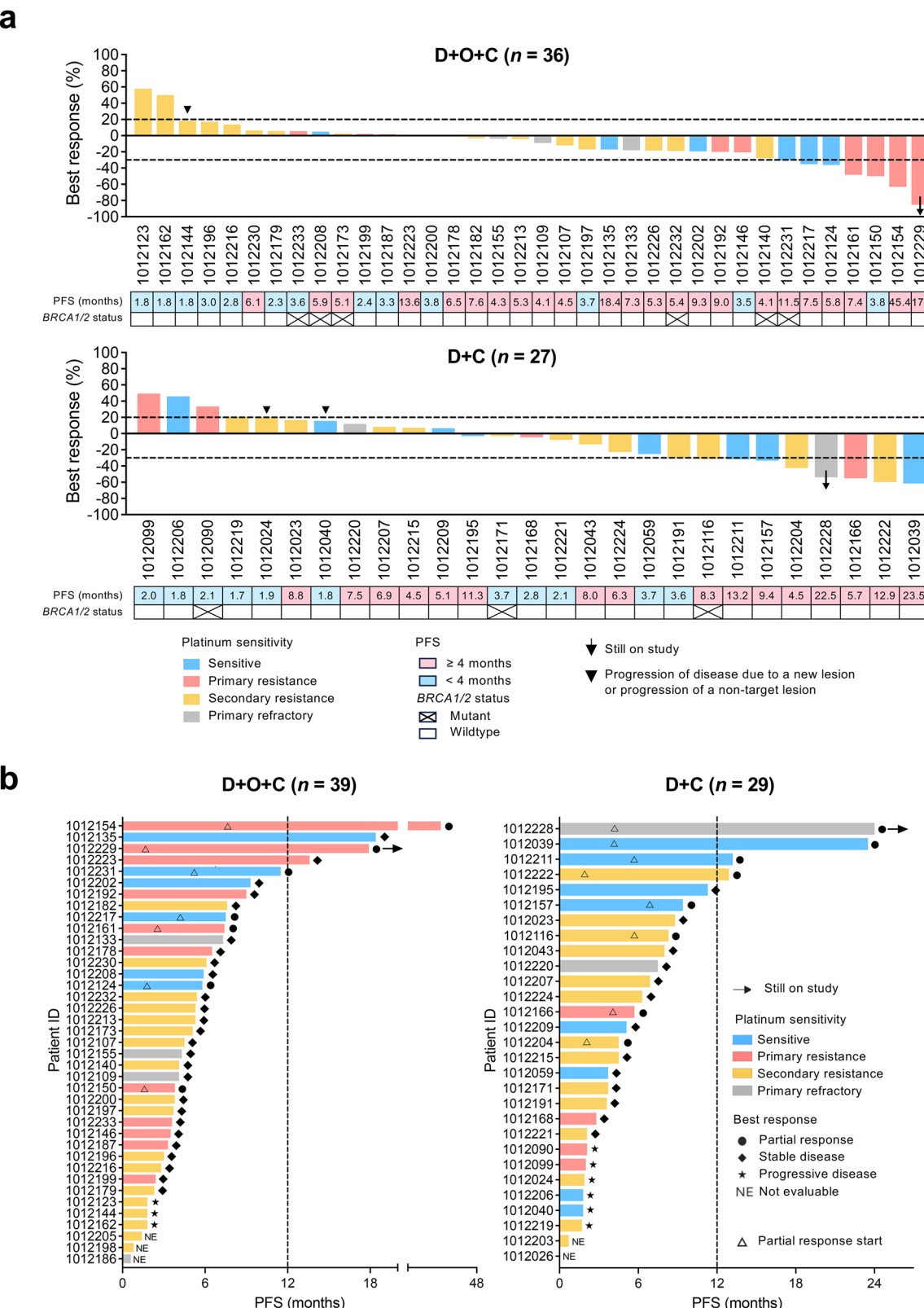

**Fig. 2 | Antitumor activity of D + O + C and D + C. a** Waterfall plots of best percent change in target lesions among RECIST-evaluable patients treated with D + O + C (*n* = 36) or D + C (*n* = 27). Dashed lines indicate thresholds for partial response (−30%) and progressive disease (+20%). Arrows represent ongoing therapy at data cut-off. *BRCA1/2* mutation status and PFS duration for each patient are shown. **b** Swimmer plots showing PFS for intention-to-treat population in D + O + C (*n* = 39) and D + C (*n* = 29) arms. Bars indicate time on treatment; symbols denote best response by RECIST v1.1. Upward triangles mark the onset of PR. Exceptional responders (PFS ≥12 months) are highlighted. Source data are provided as a Source Data file. CB clinical benefit, D + O + C durvalumab, cediranib, and olaparib, D + C, durvalumab plus cediranib, NCB no clinical benefit, PFS progression-free survival, PD progression disease, PR partial response, SD stable disease, RECIST v1.1 response evaluation criteria in solid tumors version 1.1.

investigate pathways that may have contributed to treatment response. In both treatment arms, tumors from the CB group (D + O + C [$n = 21$], D + C [$n = 14$]) exhibited significant upregulation of interferon-alpha (IFN-α) response compared with tumors from no clinical benefit (NCB; PD or SD <4 months) group (adjusted $p$ [$p$adj] = 0.002 in D + O + C and 0.014 in D + C; Fig. 3a and Supplementary Data 3 and 4). The CB group from both treatment arms also showed enrichment of multiple metabolism-related pathways (all $p$adj <0.01, Fig. 3a). Of note, no individual gene was significantly associated with clinical outcome in either treatment arm based on the differential gene expression analysis after multiple testing adjustment ($p$adj = 0.77–1; Supplementary Data 5 and 6).

### Distinct immune and metabolic signatures in exceptional responders

In a post hoc exploratory analysis, we investigated whether this pattern was also found in tumors of exceptional responders. Compared to baseline transcriptomes of the NCB group ($n = 10$), D + O + C exceptional responders ($n = 4$) showed enrichment of immune pathways (all $p$adj <0.05) but without accompanying metabolic activation (Fig. 3b and Supplementary Data 7). In contrast, tumors from D + C exceptional responders ($n = 3$) had concurrent enrichment of immune (all $p$adj <0.05) and metabolic (all $p$adj <0.01) pathways relative to NCB tumors ($n = 10$; Fig. 3b and Supplementary Data 8).

Next, we used a post hoc exploratory single-sample scoring analysis (singscore[27]) to quantify immune (Supplementary Data 9 and 10) or metabolic activity (Supplementary Data 11 and 12) in individual tumors. In the D + O + C arm, tumors from CB patients, including exceptional responders, had significantly higher immune scores than NCB tumors (Wilcoxon $p = 0.02$; Fig. 3c), and immune scores showed a trend and modest correlation with PFS (Spearman $\rho = 0.35$, $p = 0.05$; Fig. 3d). Exceptional responders tended to exhibit higher immune scores within the CB group. Metabolic scores showed no association with benefit or PFS in D + O + C (Fig. 3e, f). In contrast, in the D + C arm, CB tumors had higher metabolic scores than NCB tumors (Wilcoxon $p = 0.018$; Fig. 3e), and metabolic scores showed a modest correlation with PFS (Spearman $\rho = 0.40$, $p = 0.05$; Fig. 3f). Immune scores were not associated with benefit in D + C (Fig. 3c, d). While exceptional responders again had the highest metabolic scores, these correlations appear to be driven in part by these top-performing cases.

### Immune phenotype is also identified in an independent clinical trial dataset

In a post hoc exploratory analysis of an independent dataset from ref. 28, we examined pretreatment RNAseq from a phase II trial of ICI (pembrolizumab), VEGF blockade (bevacizumab), and DNA damage response inhibitor (metronomic cyclophosphamide) combination therapy for ovarian cancer (GSE206422; NCT02853318[28]). Using the same CB ($n = 21$)/NCB ($n = 6$) definition as in this study, we found CB tumors had higher immune scores (Wilcoxon $p = 0.03$; Fig. 3c), which positively correlated with PFS (Spearman $\rho = 0.45$, $p = 0.017$; Fig. 3d) in response to pembrolizumab+bevacizumab+metronomic cyclophosphamide, suggesting that the immune signature may have predictive relevance across the datasets. In contrast, metabolic scores showed no significant correlation with CB or PFS in this external cohort (Fig. 3e, f), consistent with our findings from the in-house data.

### Genomic profiling reveals no association between somatic alterations and treatment response

We next asked whether genomic alterations at baseline were associated with treatment outcomes. In a prespecified whole-exome sequencing (WES) correlative analysis of pretreatment tumor biopsies, *TP53* mutations were identified in the majority of cases in both D + O + C and D + C arms (Supplementary Fig. 1a, b). We examined DNA repair-related pathways and tumor mutational burden (TMB) because

homologous recombination deficiency (HRD) is linked to elevated immune activity[29], and high TMB is also associated with improved survival and increased CD8+ T-cell infiltration in ovarian cancer[30]. TMB was uniformly low (<10 mutations/Mb) for all patients across the treatment arms (Supplementary Fig. 1). In D + O + C, a CB patient carried a pathogenic *MSH2* nonsense mutation (Supplementary Fig. 1, top). One NCB patient in D + O + C carried a pathogenic *PIK3CA* missense mutation, and one NCB patient had a *KRAS* nonsense mutation in D + C (Supplementary Fig. 1). Overall, no consistent associations between baseline genomic alterations and CB were observed.

### Treatment-induced immune activation in patients with clinical benefit

Given the distinct baseline transcriptional patterns, we performed a prespecified analysis of paired pre- and on-treatment RNAseq data to assess treatment-induced transcriptional changes in the D + O + C ($n = 11$) and D + C ($n = 9$) cohorts. In the D + O + C arm, post hoc exploratory GSEA revealed that both CB ($n = 9$) and NCB ($n = 2$) tumors exhibited enrichment of immune activation pathways after treatment (Fig. 4a and Supplementary Data 13). However, only CB tumors demonstrated coordinated upregulation of specific immune activation genes, such as complement components, Fc receptors, and phagocytosis-related genes, at the transcriptional level (Fig. 4b and Supplementary Data 14). Notably, post-treatment IFN-α pathway enrichment was observed in both CB and NCB groups (Fig. 4a), whereas elevated baseline IFN-α signaling was associated with CB (Fig. 3a, b). Conversely, NCB tumors failed to show corresponding increases in individual immune effector transcripts despite pathway-level enrichment (Fig. 4b). Instead, these tumors shifted toward developmental and stromal programs, characterized by enrichment in angiogenesis and extracellular matrix (ECM) pathways (Fig. 4a and Supplementary Data 15).

In the D + C arm, CB tumors ($n = 6$) showed concurrent upregulation of immune and stromal remodeling pathways alongside downregulation of mitotic spindle and DNA repair programs, reflecting immune engagement coupled with suppressed proliferation (Fig. 4c and Supplementary Data 16). At the gene level, CB tumors exhibited increased expression of immune, angiogenesis, and ECM-related genes, while cell cycle and DNA repair genes were downregulated (Fig. 4d and Supplementary Data 17). While certain metabolic and immune pathway shifts occurred in both groups, NCB tumors ($n = 3$) uniquely displayed increased glycolysis alongside reduced fatty acid metabolism and adipogenesis (Fig. 4c). Similar to the D + O + C arm, NCB tumors in the D + C arm exhibited minimal gene-level changes in immune-related transcriptomic data despite pathway-level enrichment (Fig. 4d and Supplementary Data 18). Collectively, these data suggest that while immune pathway enrichment may be a general treatment effect, the lack of coordinated, gene-specific activation in NCB tumors may have contributed to their lack of clinical response.

### Peripheral immune cells and circulating tumor cells (CTCs)

As part of the prespecified peripheral immune monitoring, we evaluated the dynamic changes of circulating immune cells and CTCs following treatment with D + O + C ($n = 36$) or D + C ($n = 26$). Monocytic myeloid-derived suppressor cells (M-MDSC) significantly decreased by cycle 1 day 15 (C1D15) in CB groups of both arms ($p < 0.001$ in D + O + C and 0.04 in D + C), while a non-significant downward trend was observed in NCB groups ($p = 0.1055$ in D + O + C and 0.2656 in D + C; Fig. 5a). Polymorphonuclear MDSC (PMN-MDSC) levels were comparable across groups, although baseline levels were higher in NCB versus CB groups in the D + O + C arm ($p = 0.02$; Fig. 5b). Treatment with D + O + C induced systemic reduction in activated and proliferating CD8+ and CD4+ T cells across CB and NCB groups ($p < 0.05$; Fig. 5c, d) and increase in CD56 + CD16+ natural killer (NK) cells in the CB group ($p = 0.04$; Fig. 5e). Lastly, CTC levels did not demonstrate significant

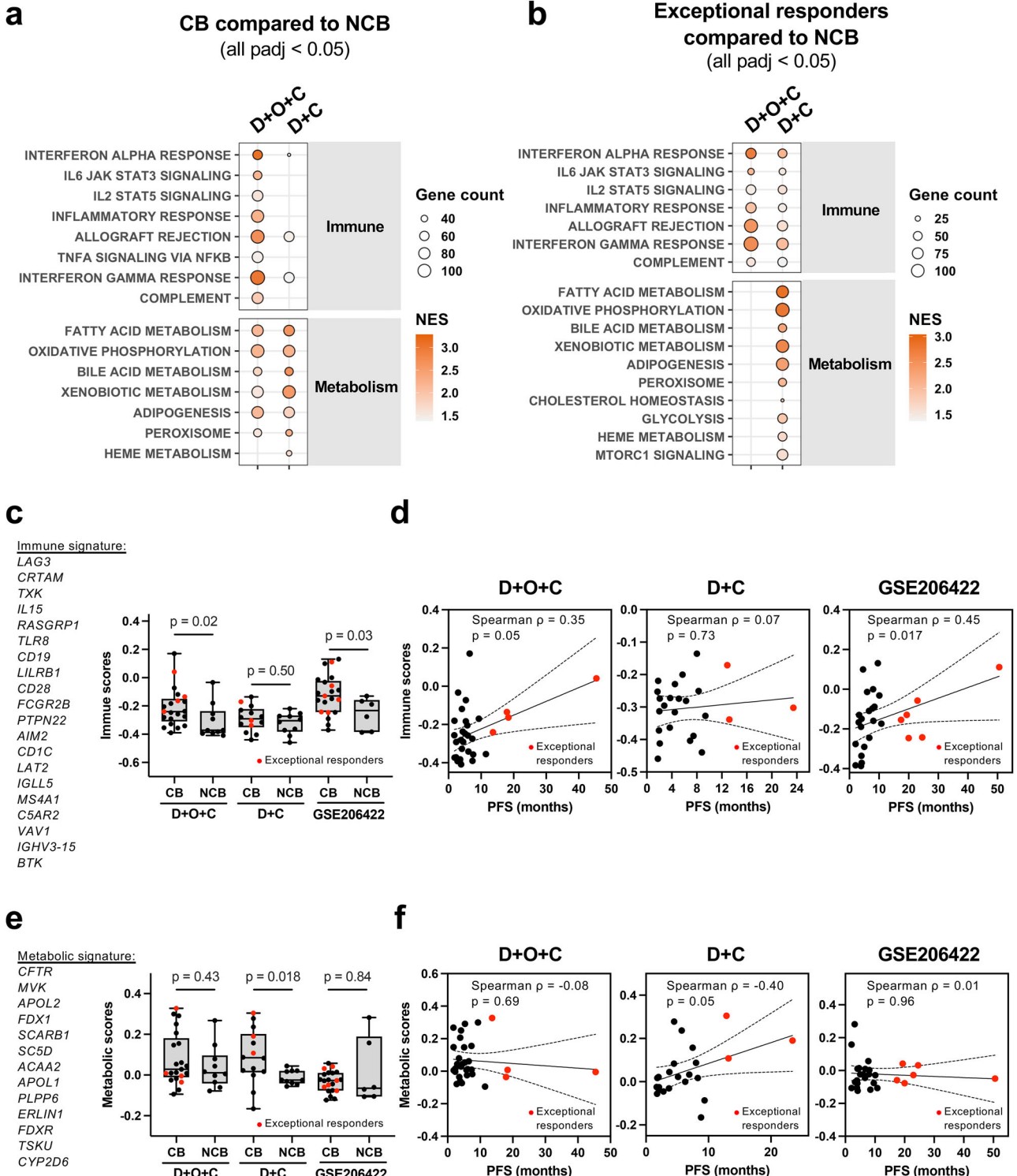

**Fig. 3 | Baseline transcriptomic characteristics associated with clinical benefit and exceptional response. a**, **b** Gene set enrichment analysis of pretreatment RNAseq comparing tumors with CB (PR or SD ≥4 months) (**a**) or exceptional response (PR or SD ≥12 months) (**b**) versus no clinical benefit (NCB: PD or SD <4 months). Pathways upregulated in CB (**a**) or exceptional response (**b**) with |NES| >1 and *p*adj <0.05 (Benjamini–Hochberg method) are shown. Circle size reflects the number of leading-edge genes. **c**–**f** Immune (**c**, **d**) and metabolic (**e**, **f**) signature scores in CB versus NCB tumors in D + O + C, D + C, and the external GSE206422 cohort. Scores in panels (**c**, **e**) were calculated using singscore, and comparisons were performed with two-sided Wilcoxon rank-sum tests. Box plots display the full range (min to max), with the median indicated by a horizontal line. Scatterplots in panels d and f show Spearman correlations with PFS. The solid line represents the linear regression fit, and the shaded area indicates the 95% CI. Pretreatment RNAseq data were available for D + O + C (21 CB, including four exceptional responders; 10 NCB; *n* = 31), D + C (14 CB, including three exceptional responders; 10 NCB; *n* = 24), and GSE206422 (21 CB, including six exceptional responders; 6 NCB; *n* = 27). Source data are provided as a Source Data file. CB clinical benefit, CI confidence interval, D + O + C durvalumab, cediranib, and olaparib, D + C durvalumab plus cediranib, NCB no clinical benefit, NES normalized enrichment score, *p*adj adjusted *p* value, PFS progression-free survival, PD progression disease, PR partial response, SD stable disease.

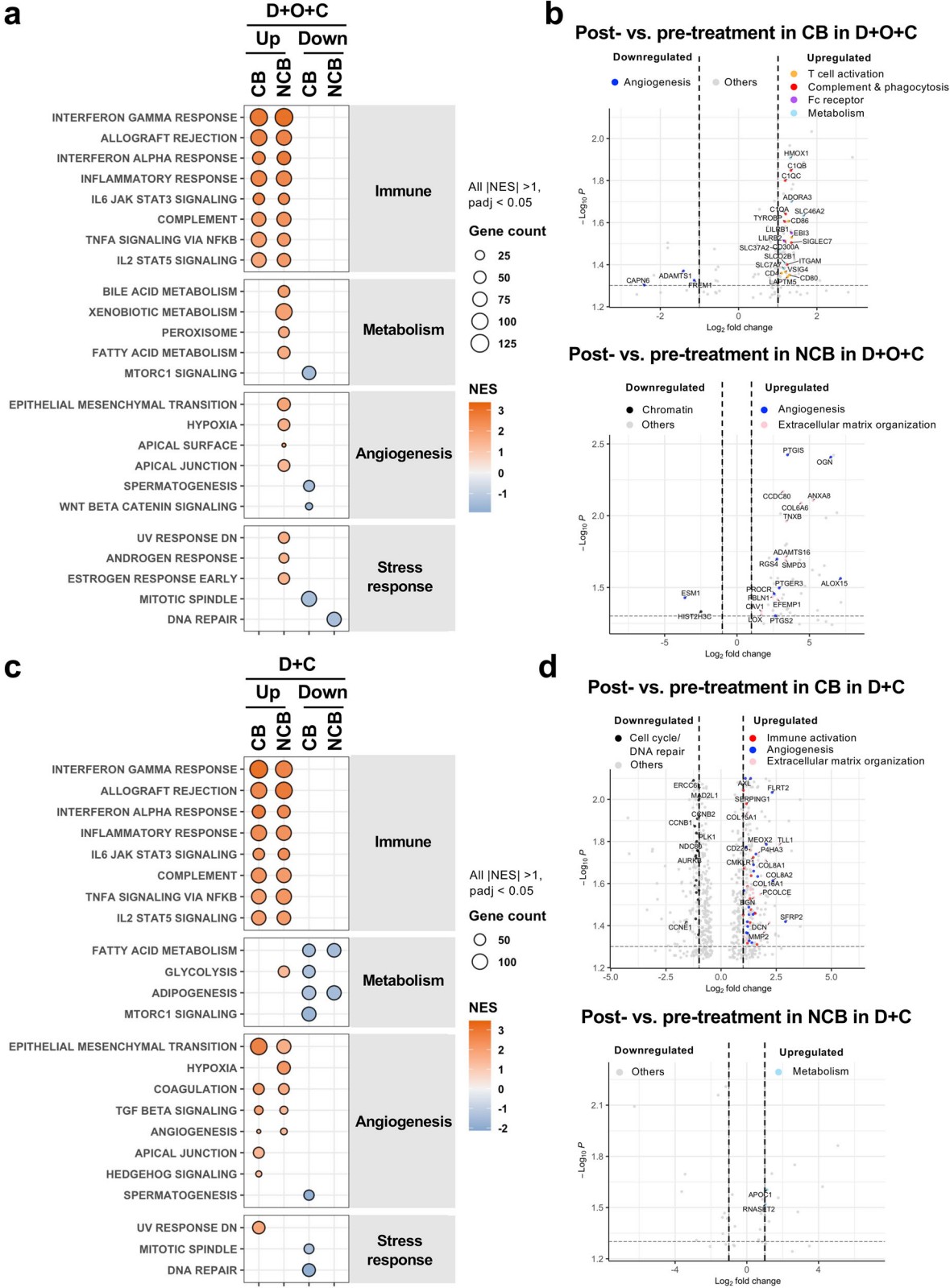

**Fig. 4 | Treatment-induced transcriptomic changes in paired tumor biopsies.** **a**, **b** In D + O + C-treated patients (CB, n = 9; NCB, n = 2), paired pre- and on-treatment RNAseq demonstrates enrichment of immune activation pathways in both CB and NCB tumors (**a**), whereas coordinated gene-level induction of immune effector transcripts occurs primarily in CB tumors (**b**). **c**, **d** In D + C-treated patients (CB, n = 6; NCB, n = 3), paired biopsies show immune and stromal remodeling in CB tumors (**c**) and increased glycolytic signaling with limited gene-level changes in NCB tumors (**d**). For panels **a**, **c**, gene set enrichment analysis was performed.

Pathways with NES >1 (upregulated) or NES <−1 (downregulated), and padj <0.05 (Benjamini−Hochberg method) are shown. Circle size reflects the number of leading-edge genes. For panels **b**, **d**, two-sided p values were calculated from the moderated t distribution. Source data are provided as a Source Data file. CB clinical benefit, D + O + C durvalumab, cediranib, and olaparib, D + C, durvalumab plus cediranib, NCB no clinical benefit; NES normalized enrichment score, padj adjusted p value, PFS progression-free survival.

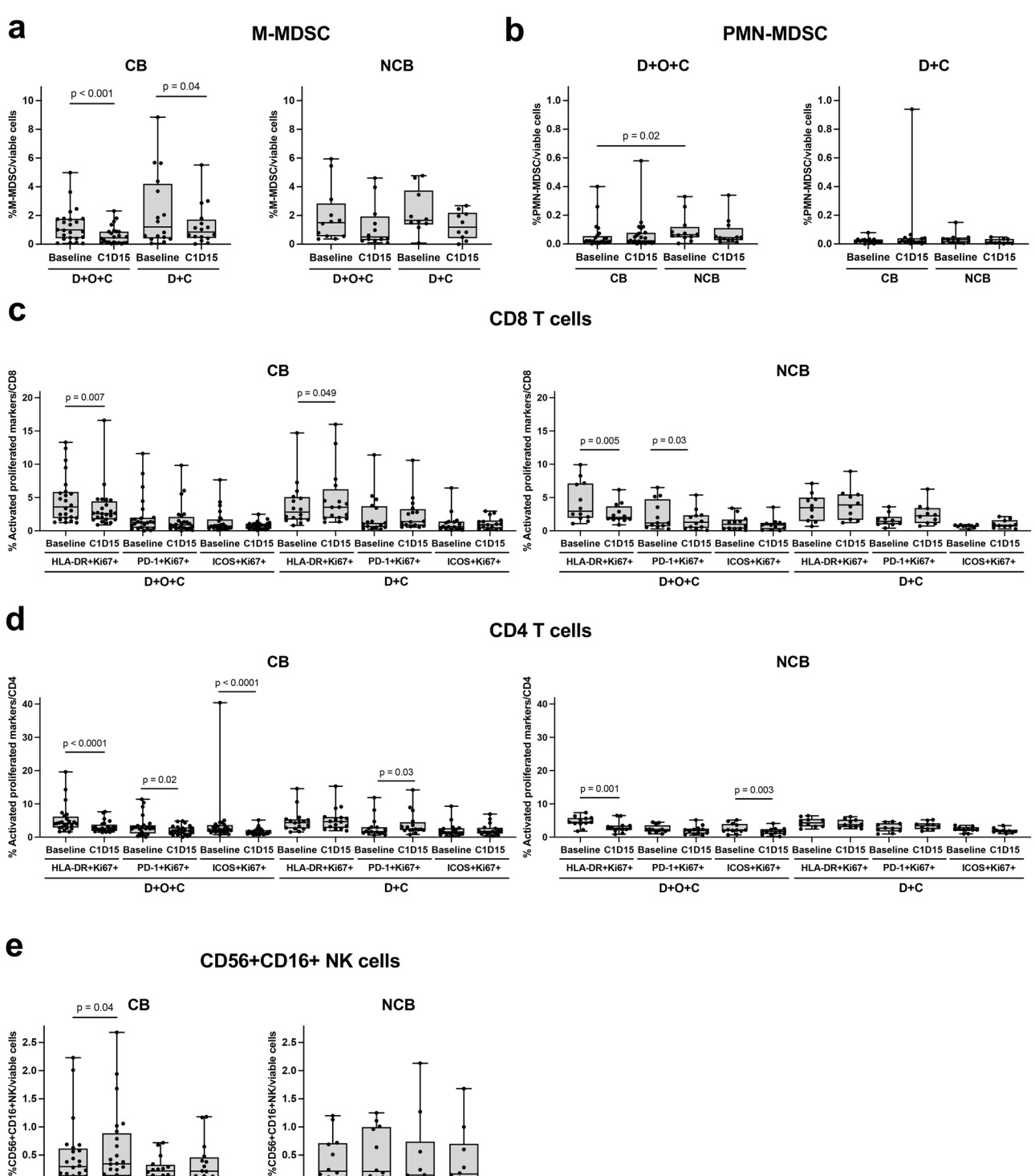

**Fig. 5 | Dynamics of circulating immune subsets following therapy.**
**a** Frequencies of monocytic M-MDSCs (CD11b+ CD14+ HLA-DR-low/− CD15−) at baseline and C1D15 in each arm. M-MDSC levels decreased at C1D15 in CB groups (D + O + C, $p < 0.001$; D + C, $p = 0.04$) but not in NCB groups (D + O + C, $p = 0.1055$; D + C, $p = 0.2656$). **b** Frequencies of PMN-MDSCs (CD14− CD11b+ CD15+) at baseline and C1D15 in CB versus NCB patients for each arm. **c, d** Activated and proliferating CD8+ (**c**) and CD4+ (**d**) T-cell subsets co-expressing Ki67 with HLA-DR, PD-1, or ICOS at baseline and C1D15 in each arm. D + O + C treatment reduced activated and proliferating CD8+ and CD4+ T cells in both CB and NCB groups ($p < 0.05$). **e** Frequencies of CD56+ CD16+ NK-cell subsets at baseline and C1D15. Sample size for **a**–**e**: D + O + C (CB, $n = 24$; NCB, $n = 12$), D + C (CB, $n = 16$; NCB, $n = 10$). The boxes extend from min to max values, with the median depicted by a horizontal line. For all data, a non-parametric two-sided Wilcoxon rank-sum test was used for unpaired samples, while a two-sided Wilcoxon matched-pairs test was used to analyze paired samples. Source data are provided as a Source Data file. C1D15, cycle 1 day 15; CB clinical benefit, D + O + C durvalumab, cediranib, and olaparib, D + C durvalumab plus cediranib, HLA-DR human leukocyte antigen-DR isotype, ICOS inducible T-cell costimulatory, M-MDSC monocytic myeloid-derived suppressor cells, NCB no clinical benefit, NES normalized enrichment score, PD-1 programmed cell death protein 1, PMN-MDSC polymorphonuclear myeloid-derived suppressor cells, NK natural killer.

changes over time and were not associated with PFS (Supplementary Fig. 2a, b).

### An 18-gene NCB signature shared across treatment arms reflects vascular remodeling and cytoskeletal plasticity and is associated with clinical outcome

To develop biomarkers identifying patients at risk of treatment failure, we compared baseline transcriptomes of NCB versus CB tumors. In NCB tumors, post hoc exploratory GSEA revealed upregulation of developmental morphogenesis and vascular adaptation signatures (all $p$adj <0.05; Fig. 6a and Supplementary Data 3 and 4). These included VEGF-driven angiogenic processes alongside non-VEGF morphogenic programs, including WNT/β-catenin, TGF-β, Notch, and Hedgehog signaling and epithelial-mesenchymal transition ($p$adj <0.0001).

NCB tumors also showed upregulation of cytoskeletal organization and microtubule dynamics pathways (Fig. 6a and Supplementary Data 3 and 4), indicating enhanced motility and cellular plasticity[31]. Arm-specific enrichment included DNA repair, cell cycle, and replication stress response pathways in D + O + C, and angiogenesis and actin dynamics pathways in D + C (all $p$adj <0.05, Fig. 6a and Supplementary Data 3 and 4).

In a post hoc exploratory effort to derive a compact resistance biomarker, we intersected the leading-edge genes from the cell morphogenesis involved in differentiation and microtubule cytoskeleton organization pathways recurrently enriched in NCB tumors across both the D + O + C and D + C arms (Fig. 6a and Supplementary Data 19). This yielded an 18-gene NCB signature (Supplementary Data 20). NCB scores were significantly higher in NCB than in CB tumors within each arm (all $p < 0.05$; Fig. 6b), and a similar trend was observed in an external dataset from patients treated with pembrolizumab, bevacizumab, and metronomic cyclophosphamide (GSE206422, Fig. 6b). Higher NCB scores correlated with shorter PFS in both arms, and the external dataset ($ρ ≤ -0.4$, $p < 0.05$; Fig. 6c), and patients with high NCB scores consistently experienced worse outcomes (Fig. 6d).

We also performed post hoc stratified analysis within the platinum-resistant subpopulation to ensure independence from prior platinum response. The predictive power of all signatures was sustained in this subgroup (Supplementary Data 21 and Supplementary Fig. 3a–c). The NCB signature achieved high accuracy in identifying NCB in platinum-resistant D + C (AUC = 0.900, $p$ = 0.005) and D + O + C (AUC = 0.806, $p$ = 0.009), with a consistent trend in GSE206422 (AUC = 0.714, $p$ = 0.207). The immune signature maintained robust performance in predicting CB in D + O + C (AUC = 0.756, $p$ = 0.031) and GSE206422 (AUC = 0.810, $p$ = 0.033), while the metabolic signature showed strong predictive value in D + C (AUC = 0.843, $p$ = 0.019), suggesting that these signatures may retain predictive potential in platinum-resistant disease.

### Microtubule-associated protein 2 (MAP2) emerges as a recurrent NCB-associated gene and exploratory marker of resistance

*MAP2* was the only gene consistently upregulated in NCB tumors across both our cohort and the GSE206422 dataset ($p$ < 0.05, Fig. 7a). In post hoc exploratory analysis, high *MAP2* expression was associated with shorter PFS in the D + O + C arm (median PFS 4.0 vs. 6.1 months; log-rank $p$ = 0.01) and the GSE206422 dataset (median PFS 4.1 vs. 8.8 months; log-rank $p$ = 0.04; Fig. 7b). *MAP2* encodes a microtubule-associated protein involved in cytoskeleton organization[32] and has been linked to drug resistance in glioma cells[33]. Its recurrent upregulation in NCB tumors suggests that MAP2 may contribute to cytoskeletal remodeling and cellular plasticity to bypass treatment-induced stress.

To examine MAP2-associated phenotypes, we evaluated protein levels and performed siRNA-mediated knockdown (Fig. 7c, d) using ovarian cancer cell lines with varying MAP2 expression, i.e., OVCAR3 and OVCAR8 (MAP2-high) and PEO1 (MAP2-low). In cancer cell-only cultures, MAP2 depletion substantially reduced tumor cell growth across all treatment conditions in MAP2-high cell lines, with the most pronounced effects observed under D + O + C treatment (Fig. 7e). In contrast, MAP2 knockdown had minimal impact on MAP2-low PEO1 growth, which remained primarily responsive to drug exposure (Fig. 7e).

In CD8+ T-cell co-cultures, T cells contributed additional cytotoxicity beyond the tumor-intrinsic effects (Fig. 7f). MAP2 knockdown enhanced T-cell–mediated killing in MAP2-high cell lines treated with D + O + C, but not in MAP2-low PEO1 (Fig. 7f). Collectively, these results suggest that MAP2 may contribute to resistance through both tumor-intrinsic mechanisms and modulation of immune susceptibility, with the former representing the dominant effect in our in vitro models.

## Discussion

Although preclinical findings support the synergy between PARPi, VEGF blockade, and ICIs[22,23,25], recent negative data from the NRG-GY023 trial underline the difficulty of translating the laboratory findings into survival benefits in bevacizumab-pretreated patients[34]. This highlights the urgent need for biomarker-driven patient selection and mechanistic translational studies. In this multi-arm, multi-cohort phase I/II trial (NCT02484404), we aimed to address this gap by identifying distinct biological determinants that influence response and resistance to the D + O + C and D + C treatment regimens. Our integrated analysis reveals that CB is associated with baseline immune activation and a complementary immunometabolic program. In contrast, treatment resistance appears to converge on shared features involving alternative vascularization and cytoskeletal plasticity. By integrating multi-omic profiling with laboratory validation, we propose a biological framework to guide more precise therapeutic strategies for future clinical trials in relapsed EOC.

In the current study, baseline immune activation emerged as a prominent correlate of CB across both treatment arms. Transcriptomic analysis from CB or exceptional responders of the D + O + C arm displayed similar enrichment of IFN-α-related pathways, aligning with models where PARPi preferentially benefits immunologically "hot" tumors[35]. Importantly, while post-treatment IFN-α upregulation occurred broadly as a pharmacodynamic effect in both arms, only baseline IFN-α signaling was associated with benefit. This distinction suggests that pre-existing immune-milieu, rather than treatment-induced interferon responses alone, establishes the permissive micromilieu necessary for sustained antitumor immunity. In the D + C arm, we observed an additional baseline enrichment of metabolic pathways in CB and exceptional responders. We hypothesize that immune cell infiltration through vascular normalization improves oxygenation and nutrient delivery[36,37] and effective T-cell functional fitness requires intact mitochondrial and cholesterol-linked programs[38]. Therefore, a metabolically active tumor microenvironment may be essential for the immune-engaging effects of combined durvalumab and cediranib, requiring further validation.

Additionally, mechanisms of resistance converged more strongly across treatment arms than mechanisms of benefit in our study. NCB tumors exhibited enrichment of pathways linked to developmental angiogenesis and cytoskeletal plasticity, features consistent with escape from VEGF-directed pressure and immune exclusion[39,40]. The recurring activation of alternative vascularization programs, such as Wnt/β-catenin and Notch, aligns with established models of non-canonical angiogenic strategies[39–41]. Simultaneously, increased cytoskeletal remodeling likely promotes cellular motility and stromal interaction[31], creating a tumor microenvironment less permissive to immune infiltration. Our findings indicate that therapeutic resistance in recurrent EOC may be driven by a coordinated phenotypic shift toward structural adaptation and alternative vascular signaling.

Among the resistance-associated genes, *MAP2* was recurrently upregulated and associated with poor clinical outcomes. Although

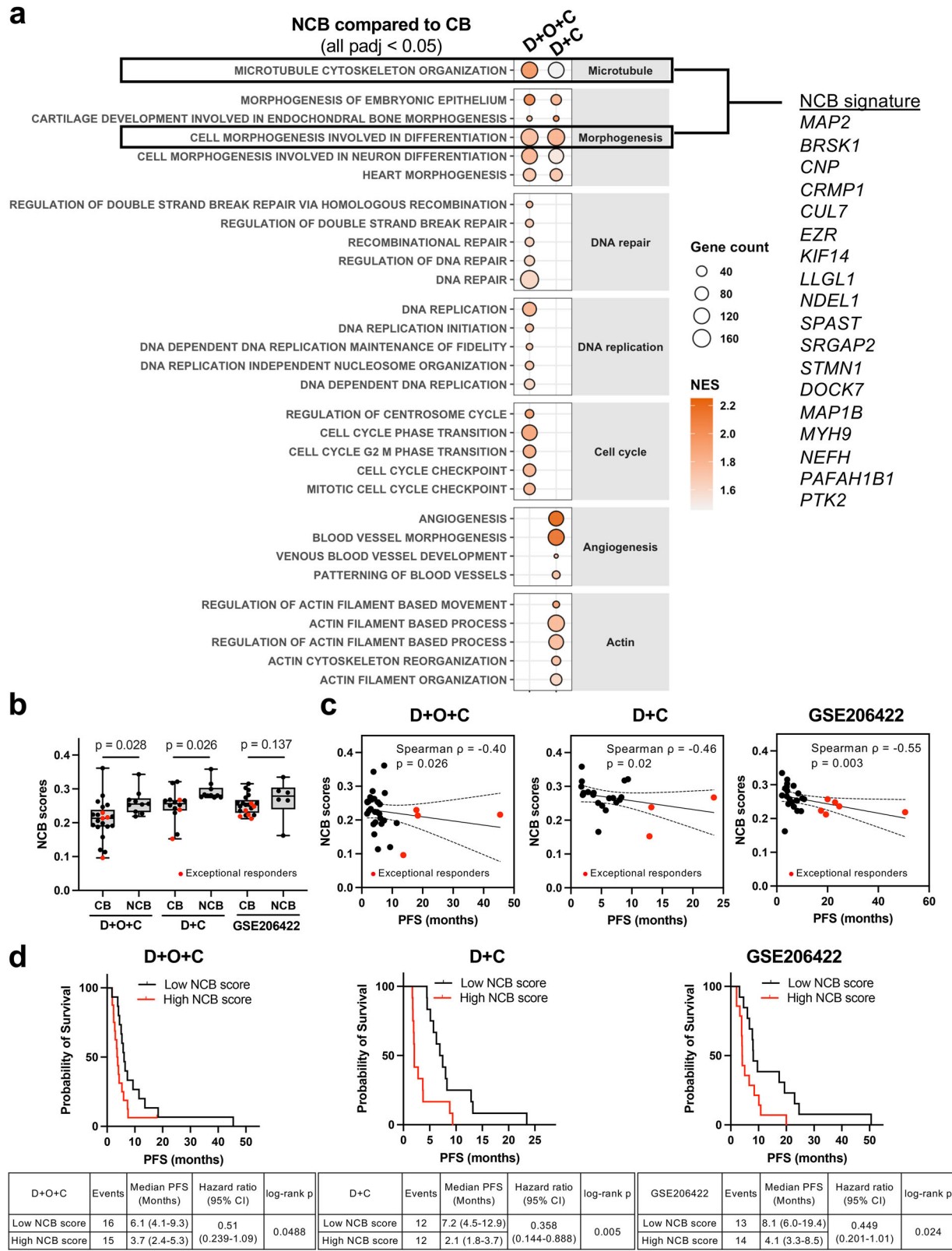

traditionally recognized as a neuronal microtubule-stabilizing protein[42], MAP2 has been implicated in cytoskeletal remodeling and drug resistance in non-neuronal cancers, including oral cancer and glioma[33,43]. Elevated *MAP2* expression in NCB tumors may reflect neuro-immune crosstalk[44] or enhanced cytoskeletal stability[45,46] that contributes to treatment resistance. In vitro, *MAP2* knockdown sensitizes ovarian cancer cells to D + O + C treatment, suggesting its

involvement in therapeutic response. From a broader clinical perspective, while the ENGOT-ov65/KEYNOTE-B96 trial[17] (pembrolizumab, bevacizumab plus paclitaxel) reported improved PFS and OS in patients with recurrent EOC, whether MAP2 serves as a predictive biomarker for such combination regimens remains undetermined. Therefore, the link between MAP2 and clinical outcomes in the context of modern combination therapies

**Fig. 6 | Transcriptomic characteristics associated with lack of clinical benefit and development of an NCB signature. a** Gene set enrichment analysis of pretreatment RNAseq identifies pathways enriched in NCB versus CB tumors across both treatment arms, including vascular adaptation, developmental morphogenesis, and cytoskeletal remodeling. Pathways upregulated in NCB with |NES| >1 and $p$adj <0.05 (Benjamini–Hochberg method) are shown. Circle size reflects leading-edge gene count. **b** NCB signature scores (18-gene overlap derived from shared cytoskeletal and morphogenesis pathways) in CB versus NCB tumors in D + O + C, D + C, and GSE206422 datasets (recurrent ovarian cancers treated with pembrolizumab, bevacizumab, and metronomic cyclophosphamide). Scores were calculated by singscore, and Wilcoxon rank-sum tests (two-sided) were used for comparisons. Box plots display the full range (min to max), with the median indicated by a horizontal line. **c** Spearman correlation between NCB scores and PFS. The solid line represents the best-fit linear regression, and the shaded area indicates the 95% CI of the regression. **d** Kaplan–Meier estimates comparing high versus low NCB score groups, with the hazard ratio, with 95% CI and log-rank $P$ value. Pretreatment RNAseq data for **a**–**d** were available for D + O + C (21 CB including four exceptional responders; 10 NCB; $n = 31$), D + C (14 CB including three exceptional responders; 10 NCB; $n = 24$), and GSE206422 (21 CB including six exceptional responders; 6 NCB; $n = 27$). Source data are provided as a Source Data file. CB clinical benefit, CI confidence interval, D + O + C durvalumab, cediranib, and olaparib, D + C durvalumab plus cediranib, NCB no clinical benefit, NES normalized enrichment score, $p$adj adjusted $p$ value, PFS progression-free survival.

should be considered hypothesis-generating and warrants further investigation.

Several factors should be considered when interpreting our findings. This study was conducted at a single center in a non-randomized setting, with a small sample size, especially in subgroup and paired biopsy analyses. Transcriptomic correlates should be considered exploratory and may be influenced by the exceptional responders. The use of bulk RNAseq limited spatial resolution for assessing immune and vascular remodeling, and scarce residual tissue prevented protein-level validation of cytoskeletal signatures, including immunohistochemical assessment of MAP2 expression. Also, we used concurrent drug-T-cell exposure for MAP2 preclinical experiments to mirror clinical conditions, thus would not exclude direct drug effects on T cells. Future studies employing sequential exposure protocols, in vivo models, and immunohistochemical validation will be essential to confirm the morphological correlates of these transcriptional findings and dissect these mechanisms. Larger prospective studies with spatial profiling are warranted.

In summary, our findings suggest that therapeutic response in recurrent EOC is more closely related to intrinsic tumor biology, including a baseline immune-milieu and metabolically active state, rather than prior platinum sensitivity or post-treatment phenotypic changes. In contrast, resistance is associated with features such as alternative vascularization and cytoskeletal remodeling at baseline, with MAP2 and the NCB signature emerging as exploratory biomarkers. Future studies that incorporate biomarker-guided selection and longitudinal profiling may help tailor treatment strategies and improve outcomes in selected patient subgroups.

## Methods
### Clinical trial
**Study design and participants.** Between September 2016 and August 2024, 68 patients were enrolled in this study. The study was conducted in accordance with the Declaration of Helsinki and Good Clinical Practice guidelines and was approved by the Institutional Review Board of the Center for Cancer Research, National Cancer Institute. All participants provided written informed consent for treatment and research use of clinical samples. This study describes the phase II EOC cohort of an open-label, multi-cohort, multi-arm, single-center phase I/II study (NCT02484404, https://clinicaltrials.gov/study/NCT02484404?tab=study, date of study registration on clinicalTrials.gov: June 29, 2015)[23,47,48]. Patients enrolled were ≥18 years of age with histologically or cytologically confirmed recurrent ovarian, fallopian tube, or primary peritoneal cancer, able to undergo fresh pretreatment core biopsies. Other eligibility criteria are detailed in the study protocol (Supplementary Information). Patients were assigned to the D + O + C or D + C arm, as per investigator's discretion based on prior treatment history.

### Clinical trial procedures
Patients in the D + O + C and D + C arms received durvalumab 1500 mg intravenously every 4 weeks, in combination with cediranib 20 mg orally once daily, administered on a 5-days-on/2-days-off schedule with or without olaparib 300 mg orally twice daily. Treatment was given in 28-day cycles and continued until disease progression, unacceptable toxicity, or withdrawal of consent. Serial blood samples were collected at baseline, C1D15, cycle 3 day 1 (C3D1), and at disease progression. Mandatory fresh core biopsies were obtained at baseline (within 24–48 h prior to C1D1), and optional on-treatment tumor biopsies prior to C1D15, C3D1 and at progression. Tumor core biopsies collected during treatment were obtained from the same lesion as the baseline biopsy whenever feasible, or otherwise from the same organ if re-sampling of the original lesion was not possible. Radiologic assessments were conducted at baseline and every two cycles (±1 week) using CT or MRI. Tumor responses were evaluated according to investigator-assessed RECIST v1.1 criteria. Patients were considered RECIST-evaluable for treatment response if they completed at least one post-treatment imaging assessment. Adverse events were recorded at each study visit and graded per Common Terminology Criteria for Adverse Events version 4.0 (CTCAE v4.0). Safety analyses included all patients who received at least one dose of study treatment.

### Study objectives and endpoints
The primary objective was to evaluate efficacy, with the primary endpoint defined as objective response rate (ORR). ORR was the proportion of patients achieving a complete response (CR) or a PR per investigator-assessed RECIST v1.1; both confirmed and unconfirmed PRs were included, with confirmation requiring a second scan performed at least 4 weeks after the initial PR. Secondary objectives included PFS, safety, and tolerability, accessed using NCI CTCAE v4.0. PFS was defined as the time from enrollment to the first documentation of disease progression (radiographic or clinical, as assessed by investigators) or death.

Prespecified translational endpoints included WES and RNAseq of tumor biopsy samples, immune cell subset profiling, and CTC analysis from peripheral blood. Post hoc exploratory analyses and mechanistic studies, including signature scoring, external dataset evaluation (GSE206422[28]), and MAP2 functional studies, were performed. Each arm was analyzed independently to identify biological correlates of response and resistance.

CB was defined as PR or SD lasting ≥4 months, aligning with the median PFS of ~4 months observed in this study. NCB was defined as PD or SD lasting <4 months. Exceptional responders (defined as PFS ≥12 months) was not prespecified in the clinical protocol and were defined retrospectively to highlight patients with markedly prolonged benefit, using the ≥12-month benchmark consistent with the NRG-GY023 trial[34].

### Samples collection
Pretreatment biopsies for genomic and transcriptomic profiling were available for 31 patients in the D + O + C arm (21 CB, including four exceptional responders, and ten NCB) and 24 in the D + C arm (14 CB, including three exceptional responders, and ten NCB); one additional exceptional responder in the D + C arm was not biopsied due to safety

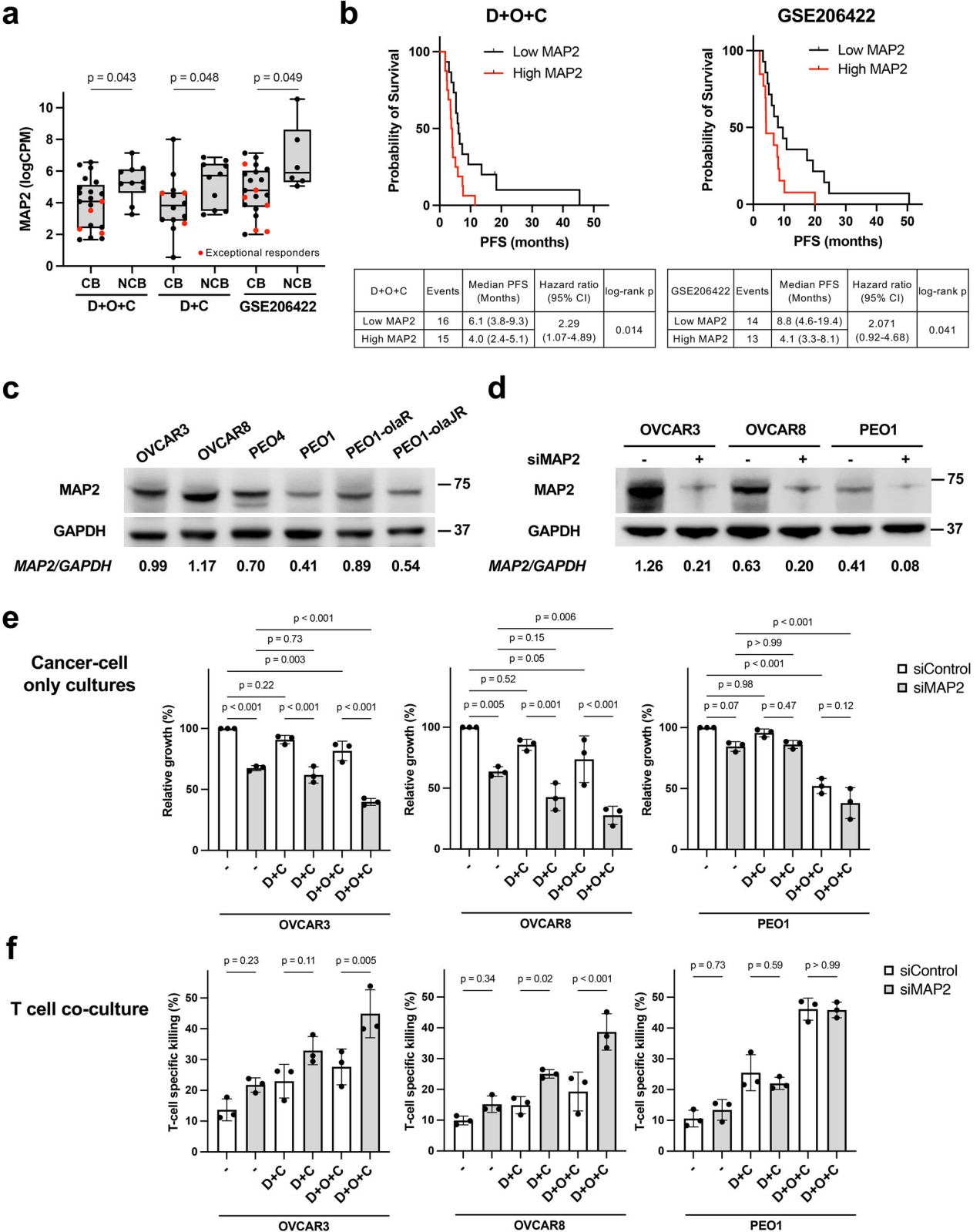

concerns. Paired pre- and on-treatment biopsies were available for 11 and nine patients in the D + O + C and D + C arms, respectively.

Biopsies were not obtained from five patients in the D + O + C arm for the following reasons: three patients were unable to undergo biopsy due to the restriction on research biopsy during COVID-19, one had tissue of insufficient quality for DNA/RNA extraction, and one

withdrew consent prior to the scheduled procedure. In the D + C arm, biopsies were not obtained from four patients: one was unable to undergo biopsy due to elevated partial thromboplastin time, two were unable to undergo biopsy due to the restriction on research biopsy during COVID-19, and one had the procedure canceled with no safely accessible target lesions on imaging.

**Fig. 7 | MAP2 expression and exploratory functional assessment of MAP2 in ovarian cancer cell line models. a** *MAP2* mRNA levels in CB versus NCB tumors from the D + O + C, D + C, and GSE206422 datasets. The boxes extend from min to max values, with the median depicted by a horizontal line. Two-sided Wilcoxon rank-sum tests were used for comparisons. **b** Kaplan–Meier curves showing shorter PFS in patients with high *MAP2* expression in D + O + C and in the external GSE206422 dataset. The PFS was assessed by a Kaplan–Meier survival plot, with the hazard ratio, with 95% CI and log-rank *p* value. *MAP2* expression data for **a**, **b** were available for D + O + C (21 CB including four exceptional responders; 10 NCB; *n* = 31), D + C (14 CB including three exceptional responders; 10 NCB; *n* = 24), and GSE206422 (21 CB including six exceptional responders; 6 NCB; *n* = 27). Wilcoxon rank-sum tests were used for comparisons. **c** Representative baseline MAP2 protein expression in ovarian cancer cell lines (*n* = 3). **d** Representative immunoblot confirmation of *MAP2* knockdown in PEO1, OVCAR3, and OVCAR8 cells (*n* = 3). **e** Side-by-side presentation of cell growth in siControl and siMAP2 cells following treatment with vehicle (DMSO + isotype IgG), D + C, or D + O + C (*n* = 3 for each cell line). *MAP2* knockdown reduced baseline proliferation most prominently in MAP2-high models (*p* < 0.001 in OVCAR3, *p* = 0.005 in OVCAR8), while drug effects were more evident in MAP2-low PEO1 cells (*p* < 0.001). **f** CD8+ T-cell co-culture assays showing changes in T-cell–mediated cytotoxicity after *MAP2* knockdown (*n* = 3 for each cell line). T-cell killing was calculated by normalizing the viability in cells with T-cell co-culture to matched those without T-cell controls. Data in panels **e**, **f** were analyzed using one-way ANOVA for multiple comparisons, and results are presented as mean ± standard deviation. Source data are provided as a Source Data file. CB clinical benefit, CI confidence interval, D + O + C durvalumab, cediranib, and olaparib, D + C, durvalumab plus cediranib, LogCPM log counts per million, MAP2 microtubule-associated protein 2, NCB no clinical benefit, PFS progression-free survival.

## WES analysis

WES was performed on genomic DNA extracted from tumor biopsy specimens and matched normal samples. DNA libraries were prepared using the Agilent SureSelect XT-Low Human All Exon V8 kit (Agilent Technologies, CA, USA) and sequenced on the Illumina NovaSeq X plus platform in 150-bp paired-end mode. Raw sequencing data were processed using the Illumina DRAGEN Bio-IT Platform (v4.2.4). Reads were aligned to the human reference genome GRCh38 (hg38). The sequencing achieved a high mean target coverage ranging from 95x to 599x, with more than 90% of the target regions covered at ≥20 x. For all samples, the Q30 score remained above 87%, and ~98% of reads were successfully mapped to the reference genome.

Somatic single-nucleotide variants and small insertions/deletions (Indels) were identified using the DRAGEN somatic pipeline in tumor-normal matched mode for 55 cases. Germline variants were identified for 55 normal samples using the DRAGEN joint genotyping mode. Tumor mutational burden (TMB) was defined as the number of non-synonymous somatic mutations per megabase of the captured exome. To achieve maximum precision in TMB normalization, the effective capture size was empirically determined by calculating the total length of the genomic intervals defined in the S33266340_Covered_hg38_v8.bed file. The precise capture size was determined to be 41.62 Mb. TMB was calculated by dividing the total count of filtered non-synonymous somatic mutations by this effective capture size.

## RNAseq analysis

RNAseq was performed on tumor biopsies to characterize gene expression profiles. RNA samples were pooled and sequenced on NovaSeq Xplus 1.5B using Illumina® Stranded Total RNA Prep with Ribo-Zero Plus and paired-end sequencing. The samples have 122 to 190 million pass filter reads with more than 92% of bases above the quality score of Q30. Reads of the samples were trimmed for adapters and low-quality bases using Cutadapt before alignment with the reference genome (hg38) and the annotated transcripts using STAR. The average mapping rate of all samples is 93%. Unique alignment is above 47%. There are 3.55 to 49.58% unmapped reads. The mapping statistics are calculated using Picard software. The samples have 4.29% ribosomal bases. Percent coding bases are between 8-50%. Percent UTR bases are 13–31%, and mRNA bases are between 21 and 77% for all the samples. Library complexity is measured in terms of unique fragments in the mapped reads using Picard's MarkDuplicate utility. The samples have 7–76% non-duplicate reads. In addition, the gene expression quantification analysis was performed for all samples using STAR/RSEM tools. Differential gene expression (DEG) analysis was conducted using the limma-voom pipeline implemented in the NIH Integrated Data Analysis Platform (NIDAP). GSEA analysis using the Molecular Signature Database (MSigDB) hallmark and gene ontology biological process (GoBP) gene set collections[49] was done by GSEA software (v4.4.0). Functional enrichment of DEG lists (Supplementary

Data 10 and 12) was performed using STRING v12.0 (https://string-db.org/).

## Single-sample gene set scoring

Single-sample gene set scoring was performed using pretreatment RNAseq data. In D + O + C, 20 genes annotated to the GO term positive regulation of immune response were selected as the immune signature (Supplementary Data 10), while in D + C, 13 genes annotated to sterol metabolic process were used as the metabolic signature (Supplementary Data 12). Shared pathways enriched in NCB tumors across both treatment arms were identified by intersecting significant gene sets. Leading-edge subsets from each arm were extracted for the overlapping pathways in cell morphogenesis and microtubule cytoskeleton organization. Intersection of leading-edge genes across arms yielded an 18-gene NCB core signature (Supplementary Data 20). We quantified the relative activity of immune and metabolic gene signatures using the singscore[27] R/Bioconductor package (v1.29.0). Resulting signature scores were merged with clinical metadata, including CB/NCB status and PFS.

In the external dataset (GSE206422[28]), raw read counts were downloaded from the Gene Expression Omnibus (GEO) and normalized using the variance-stabilizing transformation in DESeq2. Clinical data were extracted from the associated publication, including PFS and best response per RECIST. Variance-stabilized expression values for each signature gene were z-scored across samples and averaged to compute a per-sample immune signature, metabolism signature, and NCB signature scores. Samples were stratified by CB and NCB using the same thresholds as in our in-house analysis.

## Immune cell subset analysis

For immune cell subset analysis, peripheral blood samples were collected in BD Vacutainer® CPT™ Sodium Citrate tubes at baseline (pretreatment), C1D15, C3D1, and at disease progression. PBMCs were isolated from whole blood by centrifugation and viably frozen until analysis. On the day of analysis, frozen PBMCs were thawed and washed with PBS containing 0.5% BSA and 2.5 mM EDTA, followed by incubation with Fc receptor blocking reagent (#130-059-901, Miltenyi Biotec, Gaithersburg, MD, USA) used at 20 μL per $1 \times 10^7$ cells (or fewer), according to the manufacturer's instructions, and 1.0 μl LIVE/DEAD Fixable Aqua viability dye (#L34966, Life Technologies). Thereafter, samples were immunostained with monoclonal antibodies against immune subset-specific surface markers (Supplementary Data 22) for 20 min at 4 °C in the dark. Intracellular staining was performed following fixing and permeabilization using Invitrogen Fix-perm buffer (#Cat-00-5523-00). All analyses were performed using multiparametric flow cytometry (MACSQuant; Miltenyi Biotec). Data were analyzed using FlowJo software v.10.6.1 (FlowJo, LLC, OR, USA). Cells were sequentially gated on predefined immune cell subsets (Supplementary Figs. 4–6) and further characterized based on functional marker expression (Supplementary Data 23).

## CTC analysis

8 mL peripheral blood samples were collected at baseline (pretreatment), C1D15, C3D1, and at progression in BD Vacutainer® K2 EDTA tubes. Blood was centrifuged at $900 \times g$ for 7 min, and plasma was discarded. Red blood cells were lysed by incubating blood with ACK Lysing buffer (#A1049201, Thermo Fisher Scientific) for 7 min at room temperature, followed by centrifugation at $500 \times g$ for 5 min. Supernatant was discarded, and cells were washed with flow buffer (PBS with 0.5% BSA and 2.5 mM EDTA). Cell pellets were then incubated with 500 µl Hoechst 33342 (1:3000 dilution in flow buffer) (#H3570, Hoechst 33342, Life Technologies, DC, USA) for 10 min at 37 °C. After washing cells with flow buffer by centrifuging at $500 \times g$ for 5 min, cells were incubated with 1.0 µl LIVE/DEAD Fixable Aqua viability dye (#L34966, Life Technologies) and antibodies as listed in Supplementary Data 22 for 20 min at 4 °C in the dark. Following washing with flow buffer, cells were then incubated with 20 µl anti-PE magnetic beads (#130-048-801, Miltenyi Biotec) for 15 min at 4 °C in the dark to enrich EpCAM-positive cells. Cell quantification was calculated by multi-parameter flow cytometry. Viable, nucleated, EpCAM-positive, CD45-negative cells were considered as CTCs.

## In vitro study

**Ovarian cancer cell lines.** PEO1 (*BRCA2* mutation 5193 C > G, #10032308–1VL) and PEO4 (*BRCA2* reversion mutation, #10032309-1VL) were purchased from MilliporeSigma (Rockville, MD, USA). OVCAR3 and OVCAR8 (platinum-resistant *BRCA* wild-type HGSOC) were received from the NCI-60 collection at the NCI Frederick (Frederick, MD, USA). PARPi-resistant derivatives included PEO1-olaR (gift from Dr. Benjamin Bitler, University of Colorado) and in-house generated PEO1-olaJR[50]. All cell lines were cultured in RPMI-1640 with medium L-glutamine (#11875119, Life Technologies, Frederick, MD, USA) and supplemented with 10% fetal bovine serum (FBS), 1% penicillin/ streptomycin, 1 mM sodium pyruvate, and 5 µg/ml of insulin from bovine pancreas (#I0516, MilliporeSigma). PEO1-olaR was routinely maintained at 2 µM of olaparib, while PEO1-olaJR was maintained at 20 µM of olaparib. Cells were cultured without olaparib for at least 3 days prior to experiments. All cell lines were routinely tested for *Mycoplasma* using MycoAlert Mycoplasma Detection Kit (#LT-07-318, Lonza, Portsmouth, NH, USA).

## CD8+ T-cell activation for in vitro T-cell co-culture experiments

Primary human CD8+ T cells were purchased from ATCC (#PCS-800-017, Manassas, VA, USA), and cultured in RPMI-1640 supplemented with 10% FBS and recombinant human IL-2 (50 IU/mL, #200-02-50UG, Thermo Fisher Scientific, Rockville, MD, USA). Cell density was adjusted to $1 \times 10^6$ cells/mL, and the culture medium was refreshed every 2 days. On day 10, the CD8 + T cells were restimulated with CD3/CD28 Dynabeads (#11161D, Thermo Fisher Scientific) at a 1:1 bead-to-cell ratio and incubated at 37 °C, 5% $CO_2$. On day 15 post-activation, the cells were used for co-culture assays.

## siRNA transfection

ON-TARGETplus SMARTpool-Human of *MAP2* (#L-007299-00-0005) siRNAs and Dharmafect 1 reagent (#T-2001-02) were used for gene knockdown experiments as per the manufacturer's protocol (Horizon Discovery, Lafayette, CO, USA). Non-targeting control siRNAs (#D-001810-10-20, Horizon Discovery) were used as a negative control. Cells transfected with siRNA targeting *MAP2* were seeded at $5 \times 10^4$ cells per well in 24-well plates for trypan blue cell counting. Knockdown efficiency was confirmed by immunoblotting 48 h post-transfection. The sequences of siRNAs used in this study are listed in Supplementary Data 24.

## Immunoblotting

Cells were collected for protein extraction and subjected to immunoblotting. Blots were visualized using the Licor Odyssey Imaging System. MAP2 (#4542), ECL goat anti-rabbit IgG HRP (#7074) and GAPDH (#5174) antibodies were purchased from Cell Signaling Technology (Danvers, MA, USA). Antibody dilution details are provided in Supplementary Data 22.

## Drug preparation

For in vitro assays, PARPi olaparib (#S1060) was purchased from Selleck Chemicals (Houston, TX, USA). Durvalumab (#HY-P9919), cediranib (#HY-10205), and human IgG1 kappa isotype control (#HY-P99001) were from MedCahemExpress (Monmouth Junction, NJ, USA). About 100 mM of olaparib as well as 10 mM of cediranib were prepared as stocks in dimethyl sulfoxide (DMSO; #S-002-M, MilliporeSigma) and stored in aliquots at −80 °C until use.

## Cell growth assay

Cells transfected with siRNA targeting *MAP2* (siMAP2) or non-targeting control siRNA (siControl) were seeded at $5 \times 10^4$ cells per well in 24-well plates. After 24-h transfection, cells were pretreated for 24 h with cediranib (10 µM), olaparib (10 µM), durvalumab (10 µg/mL), or their combinations. DMSO (0.01% v/v) together with human IgG1 isotype (10 µg/mL) control was used as the vehicle control. For cancer cell-only assays, cell numbers were quantified after 48 h of drug exposure by trypan blue staining and manual cell counting to assess baseline drug responses.

For T-cell co-culture assays, activated CD8+ T cells were added following the 24-h drug pretreatment at an effector-to-target (E:T) ratio of 3:1 and co-cultured for 48 h in drug-containing medium without IL-2. After co-culture, non-adherent CD8+ T cells were removed by gently washing twice with PBS and collected for trypan blue staining and cell counting. T-cell−specific killing was calculated by normalizing viability in wells with T cells to the matched wells without T cells for each siRNA and drug condition.

## Statistical analyses

Each arm's design followed a Simon optimal two-stage approach[26]. The D + O + C arm was designed to test an ORR improvement from 20 to 40% (p0 = 0.20, p1 = 0.40) with α = 0.10 and β = 0.10. In the first stage, 17 patients were to be enrolled. If ≥4 responses in the first 17 patients, the second stage would enroll an additional 20 patients, with 11 or more responders of 37 patients (29.7%) would be considered positive for further development. Similarly, the D + C arm, the design aimed to rule out an ORR of 10% in favor of a target ORR of 30%, with a one-sided α = 0.10 and β = 0.10. In the first stage, 12 patients were to be enrolled. If two or more responses occurred in the first 12 patients, accrual would continue to a total of 35 patients. Six or more responses out of 35 (17.1%) would be considered positive. Safety analyses included all patients. Unfortunately, the trial was stopped early due to COVID-19, slow accrual and drug supply issues, therefore ~90% of planned enrollment was achieved.

Given this under-recruitment, we recalculated the final decision boundaries using the conditional-error approach[51], which preserved the type I error of the original two-stage design. Using the observed stage 1 responses (4/17 in D + O + C; 4/12 in D + C), we determined the minimum total number of responses required to reject the null hypothesis under the recalibrated design while maintaining the original conditional type I error. All computations were performed in R (v4.5.1) using base binomial tail probability functions (*pbinom*).

Descriptive statistics (median, frequency, and range) were used to summarize baseline characteristics, AEs, and efficacy outcomes (ORR, DCR). ORR 95% CIs were estimated using the design-consistent method of Koyama and Chen, implemented via *twostage.inference* in the *clinfun* R package[52]. Median PFS was estimated by the Kaplan−Meier method, with patients without progression censored at the last follow-up on May 13, 2025.

For RNAseq data, DEG analysis using two-sided $p$ values were derived from the moderated t distribution and adjusted for multiple testing using the Benjamini–Hochberg method ($p$adj <0.05 considered significant). GSEA results with $p$adj <0.05 and |normalized enrichment score (NES)| >1 were deemed significant. Group means of immune, metabolic, and NCB signature scores were compared using the two-sided Wilcoxon signed rank-sum test. Spearman's rank correlation was used to assess the association among immune, metabolism and NCB signature scores across patients. To ensure the signatures were independent of previous treatment history, we performed a stratified analysis by platinum sensitivity status. Predictive accuracy was assessed via the AUC. For the NCB signature, AUC was reported as 1 − $AUC_{calculated}$ to reflect its performance in predicting lack of response. Independent predictive value was further validated using multivariate logistic regression, adjusting for platinum status. AUC values >0.7 were interpreted as strong and >0.8 as excellent predictive accuracy. PFS by NCB score, *MAP2* level, or CTC change was estimated using the Kaplan–Meier method, with comparisons between arms analyzed by the log-rank test to derive hazard ratios and 95% CIs. Changes in immune cell subsets and CTCs were assessed using the two-sided Wilcoxon rank-sum test for unpaired samples and the two-sided Wilcoxon matched-pairs test for paired samples.

For in vitro studies, all experiments were performed in triplicate. Data were analyzed using one-way ANOVA for multiple comparison and are shown as mean ± standard deviation. The $p$ values <0.05 were considered significant. Statistical analyses for preclinical studies were done using GraphPad Prism v10.6.1.

### Reporting summary

Further information on research design is available in the Nature Portfolio Reporting Summary linked to this article.

## Data availability

The study protocol is available in the Supplementary Information file. Deidentified patient-level data on clinical responses are provided in the source data file. Raw clinical data are not publicly available due to data privacy regulations. Requests for access to additional deidentified clinical data should be directed to the corresponding authors. The WES data generated in this study have been deposited and are available under controlled access in the dbGaP database under accession code phs004519.v1.p1 [https://www.ncbi.nlm.nih.gov/projects/gap/cgi-bin/study.cgi?study_id=phs004519.v1.p1]. The RNAseq data generated in this study have been deposited in the GEO database under accession code GSE318308. The in-house generated PEO1-olaJR cell line is available from the corresponding authors upon reasonable request and completion of a material transfer agreement. All other data supporting the findings of this study are available within the article, the Supplementary Information, and the source data files. Source data are provided in this paper. Source data are provided with this paper.

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

## Acknowledgements

This research was funded by the Intramural Research Program of the Center for Cancer Research (CCR), National Cancer Institute (NCI), National Institutes of Health (NIH) (grant ZIA BC011525 awarded to J-M.L.). The contributions of the NIH author(s) are considered Works of the United States Government. The findings and conclusions presented in this paper are those of the author(s) and do not necessarily reflect the views of the NIH or the US Department of Health and Human Services. The authors thank T. Bao, X. Wu, A. Raziuddin, Y. Zhao, and J. Shetty at the Sequencing Facility, NCI at Frederick for performing WES and RNAseq; M. Cam at the CCR Collaborative Bioinformatics Resource, NCI for her expertise in analyzing the WES and RNAseq; C. Day at the Cancer Data Science Laboratory, NCI for his expertise in preclinical studies of immunotherapy. We also thank V. Parrish, E. Curreri, M. Gomez, S. Stearn, A. McCoy, E. Grajkowska, P. Rajagopal, T. Fujii, A. Morrill, T. Malekzandi, and C. Patel at the NCI for contributions in the clinic.

## Author contributions

Study concept and design: J.-M.L.; patients' enrollment and treatment, acquisition of data: J.-M.L., B.B.S., V.B., B.R., S. Lipkowitz, and K.C.; analysis and interpretation of data: J.-M.L., J.T., T.-T.H., N.S., S.R., J.R.N., A.B.W., S. Lee, R.L.S., and C.C.H; statistical analysis: A.Y.M., J.T., E.G., and T.-T.H.; drafting of the manuscript: J.-M.L., J.T., T.-T.H., E.G., and K.R.I.; manuscript review: all authors.

## Competing interests

J.-M.L. had research grant funding from AstraZeneca and Acrivon Therapeutics (paid to institution) and was on the Scientific Advisory Board of Acrivon Therapeutics (unpaid). Dr. Lee is currently an employee of GSK. This research work was completed prior to joining GSK and has no competing interests with GSK. The remaining authors declare no competing interests.
