## [Transparent Peer Review file · Nature Communications]

Durvalumab and cediranib with and without olaparib in recurrent ovarian cancer: a phase II proof-of-concept study

Corresponding Author: Dr Jung-Min Lee

Version 0:

Reviewer comments:

Reviewer #1

(Remarks to the Author)

The manuscript by Tabata and colleagues reports on a Phase II proof-of-concept study investigating the efficacy and mechanisms of two treatment regimens for recurrent ovarian cancer: durvalumab plus cediranib (D+C) and durvalumab, cediranib, and olaparib (D+O+C). The study enrolled 68 patients with platinum-resistant epithelial ovarian cancer (EOC) and analyzed clinical outcomes, WES, and transcriptomic data to understand treatment response and resistance mechanisms. ORR of 19.4% and 29.6% was observed in the DOC and DC arms, respectively. Based on transcriptomic analyses, the authors conclude that response is dependent on baseline immune activation in both arms and on metabolic activity in the DC arm.

Overall, the manuscript is well written and I congratulate the authors on the nicely conducted study and biomarker efforts. There are several potential concerns about the mechanistic conclusions and overinterpretation of some of the data, as outlined below.

1. Conclusions regarding correlations with PFS may be somewhat overreaching, given that correlations are not strong and appear to be driven by a small number of outliers (Fig. 3d,f)
2. The authors note that one exceptional responder in D+C group was noted to have multiple mutations in DDR genes. However, this patient appears to have ultra-mutated tumor (possibly due to POLD1/POLE mutation), where the response is more likely to be driven by high TMB. This should be noted.
3. Figure 4a,c: It doesn't appear that immune upregulation in response to therapy was any different between the CB vs. NCB groups, though the text seems to suggest that CB patients were the ones that exhibited immune upregulation. This should be reconciled. Same goes for the majority of metabolic pathways described, which seem to be going in the same direction for both CB and NCB patients (with exception of NCB patients in the DOC treatment group).
4. The conclusion that therapeutic efficacy in DC arm has a different biomarker (metabolic activity) than that in the DOC arm may be somewhat overstated. The conclusion is based on very small number of samples and is likely due to random effect rather than true biology related to the specific treatment arm (particularly since metabolic pathways were upregulated in both arms when comparing CB vs. non-CB patients and the fact that DC arm actually performed better than DOC arm). The language in abstract and manuscript claiming this should be softened. Furthermore, discussion regarding potential reasons for metabolism correlation with response is warranted. Is this biologically related to response to immunotherapy, cediranib, or both?
5. Figure 7e: The data with MAP2 knockdown are compelling, but how much is it T-cell related vs. just the intrinsic effects of drugs on the cancer cells? It doesn't appear that addition of T cells contributes much. Also, since the T cells were added to culture with drugs, one cannot exclude the possibility that cediranib directly impacts T cell function (rather than making tumor more susceptible to T cell mediated killing). This should be mentioned.

Reviewer #2

(Remarks to the Author)

Manuscript: NCOMMS-25-76416

In their manuscript, "Phase II Proof-of-Concept Study of Durvalumab and Cediranib With and Without Olaparib in Recurrent

Ovarian Cancer,” Tabata et al. present a single-center, multi-arm clinical and translational study that evaluates the efficacy of durvalumab-based combination therapies and investigates immune, metabolic, and cytoskeletal mechanisms associated with treatment response in platinum-resistant epithelial ovarian cancer.

The patient cohorts are well characterized, and the authors have assembled a unique collection of biological samples. The datasets are extensive and generally sufficient presented. However, the text is difficult to read due to poor scientific English, which limits accessibility and clarity. Although results’ significance is obvious, the manuscript needs revision before it can be recommended for publication.

The following concerns should be addressed:

General: The scientific English needs substantial refinement. Word choice—particularly verbs and adverbs—is often suboptimal, resulting in awkward phrasing and reduced readability.

1. Abstract

- a. Given that the study is positioned as a proof-of-concept, phase II investigation, the conclusion should clearly articulate whether proof of concept was achieved. As written, it highlights only the translational results, leaving the reader uncertain about the primary clinical outcome or validation of the study hypothesis.
- b. It is not immediately clear from the abstract that the data are based on an ovarian cancer cohort within a larger multi-cancer study; this should be more explicitly stated

2. Introduction

- a. After submission of this manuscript (NCOMMS-25-76416), Colombo et al. presented results from the KEYNOTE-B96 study during the Presidential Symposium at this year’s ESMO meeting, demonstrating that the addition of pembrolizumab to weekly paclitaxel, with or without bevacizumab, improved both progression-free and overall survival in patients with platinum-resistant ovarian cancer. Paragraph 3 on page 4 should be revised to contextualize the introduction in light of these new data.
- b. In paragraph 3 on page 4, “proof-of-concept” is used in a different context than in the Abstract, which may be confusing to the reader. A different term should be used to maintain consistency and clarity.

3. Methods

a. Approvals:

All necessary approvals seem to be in place.

b. Design:

The clinical protocol is well described. However, the authors do not mention the translational research endpoints, nor whether the analyses performed were preplanned or decided retrospectively.

c. Materials:

- i. Description of the cohort: This paragraph includes considerable detail that is reiterated in the Results section. Condensing this section would improve clarity and avoid redundancy.
- ii. The following information is missing from the main document
 - What types of blood samples were collected?
 - Were the different biopsies obtained from the same lesion in each patient?

d. Methods:

- i. The section would benefit from improved clarity and completeness. Many abbreviations are introduced without definition, which interrupts readability and makes it difficult to follow the methodological flow. All abbreviations (e.g., CB, NCB) should be written out in full at first mention in both the main text and figure legends.
- ii. Definition of response groups: The categories “clinical benefit (CB),” “no clinical benefit (NCB),” and “exceptional responders” should be explicitly defined in the Materials and Methods section. There is also an inconsistency in the reported number of exceptional responders (three in line 215 versus four in line 188) that should be clarified.
- iii. In the paragraph “Single-sample gene set scoring,” essential information about the analyzed samples and the analytical setup (with reference to the Supplementary Material) is missing. In the current version, it is difficult to understand which data were included in the analysis and how they were selected.
- iv. Multiple analytical approaches were applied to the genomic data. Given that these analyses form the core of the Results section, a more detailed description should be provided here to ensure clarity and context. E.g. Gene set scoring: In the paragraph describing single-sample gene set scoring, please provide details on the genes annotated to the immune and metabolic signatures (20 genes for DOC and 15 for DC, respectively; line 473). These gene lists should also be made available in the main text or shown in Figure 3 to improve transparency.
- v. The manuscript should include details of the flow cytometry panels used for immune profiling (antibody list and definition of immune subsets) and for the identification of circulating tumor cells (CTCs), to ensure methodological transparency and reproducibility.
- vi. Immune cell subset analysis: Handling of peripheral blood mononuclear cells (PBMCs) should be described in more detail. Include information on the concentrations of Fc block and live/dead cell markers, and provide a complete antibody list (clone, fluorochrome, and concentration) to ensure reproducibility.
- vii. In vitro cell growth and drug sensitivity assays (Figure 7e): Please clarify whether dimethyl sulfoxide (DMSO) was used as a vehicle control and specify its final concentration in the culture medium. The reported growth inhibition appears to be primarily associated with the MAP2 knockout rather than the treatment effect. It would also strengthen the analysis to include PEO1 cells as a positive control.

4. Results

This section should be limited to the presentation of results; interpretation should be reserved for the Discussion.

- a. The manuscript would benefit from a more balanced presentation of the data. While the genomic analyses are comprehensively detailed, the phenotypic characterization of peripheral immune cells and circulating tumor cells (CTCs), as well as the functional studies, receive minimal attention and should be expanded for completeness.
- b. Currently, some important translational observations are not fully contextualized, and key comparisons are underdeveloped.
 - i. IFN- α expression: The reported upregulation of interferon-alpha (IFN- α) in patients with clinical benefit before treatment requires further clarification. The post-treatment increase of IFN- α observed in both clinical benefit (CB) and no clinical benefit (NCB) groups suggests that this may represent a treatment-related rather than response-specific effect. The authors should reconcile this observation with the relative pretreatment elevation of IFN- α in the CB group (Figure 3a).
 - ii. Cytoskeletal and microtubule pathway activation: NCB tumors reportedly show enrichment of cytoskeletal organization and microtubule dynamics pathways (lines 311–312). The biological and morphological relevance of these findings is not talked over (in the Discussion). Were there observable histological or phenotypic differences between these tumors? Since the abstract proposes cytoskeletal profiling as a potential predictive marker for lack of clinical benefit, validation using non-transcriptional methods (e.g., morphological or immunohistochemical assessment) would strengthen this conclusion.
 - iii. MAP2-related growth effects: The analysis of microtubule-associated protein 2 (MAP2) expression in NCB tumors indicates a role in modulating microtubule dynamics (line 335), yet the rationale for the experimental design is unclear. Given that MAP2 expression is high in OVCAR3 and OVCAR8 cells, why was PEO1 not included as a positive control in the drug-response assays? The observed growth reduction across all knockout models, including untreated and vehicle-treated (dimethyl sulfoxide, DMSO) controls, suggests that the inhibitory effect may derive primarily from the knockout rather than from drug treatment. Clarification is needed to distinguish treatment-specific effects from MAP2-dependent growth suppression.

5. Figure legends and Figures

- a. Most figure legends require rephrasing for clarity and completeness. Each legend should provide a concise description of the experimental setup, sample size, and main findings to enable understanding without referring to the main text.
- b. Most of the figures are well designed and effectively illustrate the findings.
 - i. Figure 1: The number of analyzed samples (blood and biopsies) should be clearly indicated within the figure itself, not only in the legend.
 - ii. Figure 2: Panels (a) and (b) appear to display identical results using different visualization methods. Please clarify the rationale for presenting both or consider consolidating them.
 - iii. Figure 3: Panels (g) and (h) are missing—see the figure legend. The Spearman correlation analysis is not convincing (e.g., line 222 $p = 0.35$; line 227 $p = 0.40$), indicating only weak correlations, both for immune and metabolic scores.
 - iv. Figure 4: The observed upregulation appears to reflect a treatment effect rather than a response-associated effect. Please clarify how the post-treatment upregulation of IFN- α in both CB and NCB groups aligns with the relative pretreatment increase of IFN- α in the CB group (Figure 3a).
 - v. Figure 5: Violin plots are best suited for large datasets. Given the limited sample size, boxplots are recommended for improved clarity and transparency. The number of CB and NCB samples included should be explicitly indicated. Expression markers for m-MDSCs and PMN-MDSCs (panels a and b) should be defined in the figure legend.
 - vi. Figure 7: The comparison between MAP2A-expressing and knockout (KO) cells across treatment groups would be more intuitive if shown side by side, rather than comparing treatment effects independently. This arrangement would better highlight differences attributable to MAP2A expression status.

6. Discussion

The Discussion should be more focused and structured. The first paragraph should concisely summarize the main findings. Thereafter, three to four major results should be interpreted individually and in relation to the existing literature before being placed in a broader clinical perspective. Each section should end with a clear point of interpretation rather than a preliminary conclusion. The authors should also integrate the clinical outcomes, immune profiling, and functional data into a more overarching biological and clinical context. As noted in the comments on the Introduction, the recently presented results from the KEYNOTE-B96 study provide important new clinical data that should be discussed in relation to the present findings. In addition, the translational data require validation, which should be addressed explicitly in this section.

Reviewer #3

(Remarks to the Author)

I have focused on evaluation of trial design/methodology, reporting of outcomes, and statistical analyses methods.

Abstract

Translational results are highlighted without any data reported - which of these analyses were pre-specified in the protocol/Analysis plan? In the discussion, the exploratory nature of the biomarker analyses is highlighted.

Methods - Study objectives and endpoints:

Section states (clinical) endpoints rather than objectives. [ie Primary objective as per clintrials.gov: To determine how effective this combination is in treating ovarian cancer. Then primary endpoint is ORR as measure of antitumour activity]. This section would benefit from specifying pre-specified secondary objectives for translational analyses (and which ones were post-hoc/exploratory).

Definition of ORR: complete (CR) or confirmed/unconfirmed PR per investigator-assessed RECIST v1.1. (how was this

confirmed? Central review? Second scan?

PFS - does it include clinical progression or was it just imaging-defined progression?

In many of the results, and also in the abstract, Clinical benefit (CB: PR or SD>4m) or exceptional responders (PFS>12m) are the key endpoints analysed, but these are not defined in objectives/endpoints (nor are in clinicaltrials.gov). Was this endpoint/objective pre-specified in the trial? Is the duration of SD 'standard' definition for CB?

Methods - Statistical analyses

Exact 95% confidence intervals are used for the ORR - but these are not consistent with the 2-stage design. Better inference methods have been advocated: <https://pmc.ncbi.nlm.nih.gov/articles/PMC6047527/>
And are include in R packages, such as `clinfun` (function `twostage.inference`).

Similarly, there are methods for proper handling of under-recruitment in phase 2 studies, which allow to 'recalculate' decision thresholds in the 2-stage design: <https://pubmed.ncbi.nlm.nih.gov/25781860/> (I think also implemented in the above R package).

I am unsure by the attempt to demonstrate statistical equivalence testing between the two treatment arms. This is not an aim/stated in the protocol. I understand the aim of this analysis was to enable the "pooled" analysis of the two cohorts for biomarker analyses. But as designed, it was never aimed at comparing these two treatment groups, patients were not randomised to one or the other, patients had, a priori, different histologies (and selection bias to be allocated into each arm). Was this a pre-specified analysis? Importantly, was the equivalence margin pre-specified a priori? FYI, there are equivalence methods for censored data (but maybe not in the TOSTER package) although I accept with the low % censoring it does not make a big difference. I'd argue that normality test for PFs is 'redundant' as time to event endpoints are usually skewed. In short, I'd remove this whole bit from methods (no linked results) and state all results are presented by cohort as different populations in any case.

For the translational analyses, largely non-parametric tests were used (page 20, lines 584-589). Results indicate that time-to-event analysis were performed (figures 6d and 7b) but this is not explained in the methods. Results clearly show adjustment of multiplicity was implemented (adjusted p-values) but this is not explained in the methods. There is mention to "one-way ANOVA for multiple comparison" in line 591, but I interpreted that this corresponds to the in-vitro studies, is this correct? A reference for the ANOVA method may be useful.

Lines 591-592 "The $p < 0.05$ were considered significant. All statistical analyses were done using GraphPad Prism v10." Does this refer to the whole analysis or just the in-vitro analysis?

Results - Patient enrollment and baseline characteristics

Please report recruitment start/stop dates for each cohort separately.

Results - Efficacy and safety

Please state how many of the PR responses were unconfirmed.

When refer to patients having SD, needs to be specified this is 'best overall response as per RECIST - I think only stated in figure 1b legend. Is this something that should be more clear in the endpoint definition?

This is the main results for cohorts D+C and D+C+O but there is not a statement regarding the primary outcome of the trial - Although it under-recruited, the number of responses observed permit to conclude according to the proposed design: 7PR in 36 D+O+C (planned $\geq 11/37$)  based on ORR, D+O+C did not reach its threshold to investigate further; 8PR in 27 D+C (planned $\geq 6/35$), so even if recruitment had continued, conclusion would have been that D+C is promising to investigate further (based on ORR).

Results - Upregulation of immune and metabolic pathways

I would be explicit in stating "In both treatment arms, tumors from CB group (D+O+C [n = 21], D+C [n = 14]) exhibited significant upregulation of interferon alpha response than tumours in NCB group (all adjusted p...)"

Reviewer #4

(Remarks to the Author)

Version 1:

Reviewer comments:

Reviewer #1

(Remarks to the Author)

I'm satisfied with author responses to my comments. I have no further comments.

Reviewer #2

(Remarks to the Author)
NCOMMS-25-76416A

The manuscript has improved substantially following revision. The majority of our previous comments have been addressed, and the necessary modifications have been implemented. Overall, the revisions have improved the clarity, coherence, and scientific presentation of the study.

Major comment:

It remains unclear why the authors chose to compare only the D+O+C arm with the D+C arm, while excluding the D+O arm—which, based on the rationale presented in the Introduction, would appear to be a more logical comparator—or why all three treatment arms were not included in the comparative analyses. This issue requires clearer justification and should be addressed consistently throughout the different sections.

Minor comments:

Abstract

- The cohorts should be presented in the same order throughout the paragraph.
- The last sentence should be rephrased. Suggested revision: "These findings support proof-of-concept clinical activity of D+O+C and D+C and identify molecular signatures with potential predictive value in subsets of recurrent EOC."

Introduction
OK

Methods

- Please rephrase the description of the translational research end-points to better reflect the results from analyses performed and the results obtained (page 16, lines 454–457).
- Given that only 13 genes are annotated to the metabolic signature (Supplementary Table 11 and Figure 3e), the current description in the Methods section requires revision
- Immune cell subset analysis: When describing the handling of peripheral blood mononuclear cells (PBMCs), the concentration of the Fc-blocking agent could be reported rather than the volume used (page 17).

Results

- Main document: The Figures have been included twice.
- The following statement "The concomitant upregulation of angiogenesis and ECM genes, alongside downregulation of chromatin and endothelial transcripts, suggests a state of immune exclusion with enhanced tissue plasticity" (page 10) could be removed and eventually moved to the Discussion.

Figure legends and Figures

- Figure 6a: Add "NCB" to improve clarity.
- Significant IFN- α upregulation: Include the p-values to the text (page 7).
- Figure 5: The p-values for the NCB group in the M-MDSC analyses should be reported in the text, as a trend toward M-MDSC downregulation at C1D15 is also observed in this group. In addition, Figure 5f is not described in the Results section; if this panel is redundant, it should be removed from Figure 5 (for example, replaced with the CTC results shown in Extended Figure 2a/b).

Discussion

- The relevance of the following statement is unclear: "Our data showing that MAP2 knockdown sensitizes cells to treatment may be clinically relevant given the KEYNOTE-B96/ENGOT-ov65 trial4 (pembrolizumab, bevacizumab plus paclitaxel) reported improved PFS and OS in patients with recurrent EOC, although this requires further mechanistic studies and prospective clinical validation." The mechanistic link between the presented preclinical findings and the clinical outcomes reported in this trial is not sufficiently explained and requires clearer justification.

Others

- Clinical benefit (CB) and no clinical benefit (NCB) should be defined at first mention.

Reviewer #3

(Remarks to the Author)
Thank you - the authors have addressed my comments and concerns.

Reviewer #4

(Remarks to the Author)

Version 2:

Reviewer comments:

Reviewer #2

(Remarks to the Author)

The authors have addressed all comments and concerns.

Reviewer #4

(Remarks to the Author)

Reviewer #1

The manuscript by Tabata and colleagues reports on a Phase II proof-of-concept study investigating the efficacy and mechanisms of two treatment regimens for recurrent ovarian cancer: durvalumab plus cediranib (D+C) and durvalumab, cediranib, and olaparib (D+O+C). The study enrolled 68 patients with platinum-resistant epithelial ovarian cancer (EOC) and analyzed clinical outcomes, WES, and transcriptomic data to understand treatment response and resistance mechanisms. ORR of 19.4% and 29.6% was observed in the DOC and DC arms, respectively. Based on transcriptomic analyses, the authors conclude that response is dependent on baseline immune activation in both arms and on metabolic activity in the DC arm.

Overall, the manuscript is well written and I congratulate the authors on the nicely conducted study and biomarker efforts. There are several potential concerns about the mechanistic conclusions and overinterpretation of some of the data, as outlined below.

1. Conclusions regarding correlations with PFS may be somewhat overreaching, given that correlations are not strong and appear to be driven by a small number of outliers (Fig. 3d,f)

Response: We agree with the reviewer that the correlations with PFS may have been driven by the limited sample size and exceptional responders, requiring interpretation with caution and prospective validation in large cohorts. We revised the Results (**page 8**) and Discussion (**page 14**) to reflect this concern.

2. The authors note that one exceptional responder in D+C group was noted to have multiple mutations in DDR genes. However, this patient appears to have ultra-mutated tumor (possibly due to POLD1/POLE mutation), where the response is more likely to be driven by high TMB. This should be noted.

Response: We thank the reviewer for raising this important point. To address this, we first cross-checked a high TMB from our inhouse data (non-CLIA-certified lab) with a CLIA-certified NGS report, performed by Caris Life Sciences, which showed a low TMB of 6 mutations/Mb. This discrepancy led us to re-investigate the whole datasets including WES to recheck the accuracy.

Our investigation identified that the data were generated from tumor only WES, not matched to the normal PBMC samples for each patient, although we were initially informed by the NCI Frederick Sequencing and Genomics Core and Bioinformatics Core that they were tumor-normal sample matched WES analyses.

We apologize for the oversight and believe that our initial report on *POLE/POLD1* may have been overcalled polymerase gene variants given the absence of matched normal sequencing. After re-analysis using tumor-normal matched WES, we confirmed that this exceptional responder in the D+C arm didn't harbor pathogenic *POLE* or *POLD1* mutations and TMB was 1.8 mutations/Mb.

We have updated the data (**Extended Data Fig. 1**) and revised the manuscript accordingly (**page 9**).

3. Figure 4a,c: It doesn't appear that immune upregulation in response to therapy was any different between the CB vs. NCB groups, though the text seems to suggest that CB patients were the ones that exhibited immune upregulation. This should be reconciled. Same goes for the majority of metabolic pathways described, which seem to be going in the same direction for both CB and NCB patients (with exception of NCB patients in the DOC treatment group).

Response: We thank the reviewer for identifying this inconsistency and have updated the manuscript (**pages 9-10**). Briefly, we specified that while pathway-level enrichment was observed in both groups, only CB tumors demonstrated coordinated upregulation of immune effector genes at the individual gene level. In contrast, NCB tumors lacked meaningful gene-level immune activation and instead shifted toward developmental and stromal programs, marked by increased angiogenesis and extracellular matrix gene expression. Also, we clarified that enhanced glycolytic programs were uniquely present in NCB tumors within the D+C arm.

4a. The conclusion that therapeutic efficacy in DC arm has a different biomarker (metabolic activity) than that in the DOC arm may be somewhat overstated. The conclusion is based on very small number of samples and is likely due to random effect rather than true biology related to the specific treatment arm (particularly since metabolic pathways were upregulated in both arms when comparing CB vs. non-CB patients and the fact that DC arm

actually performed better than DOC arm). The language in abstract and manuscript claiming this should be softened.

Response: We have updated the Abstract (**page 3**) and Results (**page 8**) to describe these metabolic associations as exploratory findings, warranting further investigation.

4b. Furthermore, discussion regarding potential reasons for metabolism correlation with response is warranted. Is this biologically related to response to immunotherapy, cediranib, or both?

Response: Thank you for your comment. We initially hypothesized that improved tumor oxygenation and nutrient delivery by VEGFR inhibition^{1,2} would activate mitochondrial and cholesterol-linked metabolic programs³ which are essential for T-cell functional fitness. Therefore, a metabolically active tumor environment may facilitate immune engagement by combined durvalumab and cediranib. However, we agree with the reviewer that this metabolism correlation with response could be related to both drugs or either alone. To test this hypothesis, proof-of-concept clinical trial and translational studies should be conducted to study this important question. We have expanded the Discussion (**page 13**) to acknowledge that further clinical and translational studies are needed to address the biological basis.

5. Figure 7e: The data with MAP2 knockdown are compelling, but how much is it T-cell related vs. just the intrinsic effects of drugs on the cancer cells? It doesn't appear that addition of T cells contributes much. Also, since the T cells were added to culture with drugs, one cannot exclude the possibility that cediranib directly impacts T cell function (rather than making tumor more susceptible to T cell mediated killing). This should be mentioned.

Response: We appreciate this important observation. To dissect the relative contributions of tumor-intrinsic versus T-cell-mediated effects, we have performed new parallel experiments across MAP2-high and MAP2-low cell lines, comparing cancer cell-only cultures with CD8⁺ T-cell co-cultures.

Briefly, in cancer cell-only cultures, MAP2 knockdown substantially reduced tumor cell growth across all treatment conditions in MAP2-high models (OVCAR3 and OVCAR8), with the most pronounced effects under D+O+C treatment (**new Fig. 7e**). In contrast, MAP2 depletion had

minimal impact on MAP2-low cell line (PEO1), where growth inhibition remained primarily drug-driven (**new Fig. 7e**). These data indicate a substantial tumor-intrinsic component of MAP2-mediated resistance.

Additionally, further cytotoxicity was observed upon addition of CD8⁺ T cells (**new Fig. 7f**). MAP2 knockdown in MAP2-high cell lines treated with D+O+C, enhanced T-cell-mediated killing (e.g., ~45% in OVCAR3 siMAP2 vs. ~27% in siControl), whereas MAP2-low PEO1 remained predominantly drug-responsive with negligible MAP2-dependent enhancement (**new Fig. 7f**). These findings indicate that while tumor-intrinsic mechanisms dominate, T cells provide measurable additional killing in MAP2-high contexts.

Lastly, we agree with the reviewer that our experimental design wouldn't exclude the possibility of cediranib's direct effects on T-cell function. Our intention was to mirror the clinical scenario for the preclinical experimental design where immune cells and therapeutics coexist. This limitation is now acknowledged in the Discussion (**page 14**).

Reviewer #2

Manuscript: NCOMMS-25-76416

In their manuscript, “Phase II Proof-of-Concept Study of Durvalumab and Cediranib With and Without Olaparib in Recurrent Ovarian Cancer,” Tabata et al. present a single-center, multi-arm clinical and translational study that evaluates the efficacy of durvalumab-based combination therapies and investigates immune, metabolic, and cytoskeletal mechanisms associated with treatment response in platinum-resistant epithelial ovarian cancer. The patient cohorts are well characterized, and the authors have assembled a unique collection of biological samples. The datasets are extensive and generally sufficient presented. However, the text is difficult to read due to poor scientific English, which limits accessibility and clarity. Although results' significance is obvious, the manuscript needs revision before it can be recommended for publication.

The following concerns should be addressed:

General: The scientific English needs substantial refinement. Word choice—particularly verbs and adverbs—is often suboptimal, resulting in awkward phrasing and reduced readability.

Response: We have carefully edited the manuscript for clarity and readability.

1. Abstract

1a. Given that the study is positioned as a proof-of-concept, phase II investigation, the conclusion should clearly articulate whether proof of concept was achieved. As written, it highlights only the translational results, leaving the reader uncertain about the primary clinical outcome or validation of the study hypothesis.

Response: We agree and have revised the Abstract to explicitly state the proof-of-concept outcome.

1b. It is not immediately clear from the abstract that the data are based on an ovarian cancer cohort within a larger multi-cancer study; this should be more explicitly stated

Response: We have stated that this is a recurrent ovarian cancer cohort in a phase I/II multi-treatment arm, multi-disease cohort trial.

2. Introduction

2a. After submission of this manuscript (NCOMMS-25-76416), Colombo et al. presented results from the KEYNOTE-B96 study during the Presidential Symposium at this year's ESMO meeting, demonstrating that the addition of pembrolizumab to weekly paclitaxel, with or without bevacizumab, improved both progression-free and overall survival in patients with platinum-resistant ovarian cancer. Paragraph 3 on page 4 should be revised to contextualize the introduction in light of these new data.

Response: We thank the reviewer for highlighting the newly presented KEYNOTE-B96 results⁴. We have incorporated the KEYNOTE-B96/ENGOT-ov65 results into the Introduction (**pages 4-5**), highlighting the improvement in PFS and OS with pembrolizumab plus weekly paclitaxel ± bevacizumab in platinum-resistant EOC. We contextualize our study as exploring non-chemotherapy-based ICI combinations that may provide alternative treatment options complementary to this chemotherapy-based strategy.

2b. In paragraph 3 on page 4, “proof-of-concept” is used in a different context than in the Abstract, which may be confusing to the reader. A different term should be used to maintain consistency and clarity.

Response: We removed the term “proof-of-concept” when referring to preclinical studies to keep its usage consistent with the clinical context.

3. Methods

3a. Approvals: All necessary approvals seem to be in place.

Response: Thank you.

3b. Design: The clinical protocol is well described. However, the authors do not mention the translational research endpoints, nor whether the analyses performed were preplanned or decided retrospectively.

Response: We have specified in the Methods (**page 16**) that RNAseq, WES, immune subset profiling, and CTC analyses were preplanned post-hoc translational endpoints in the original study protocol, whereas signature scoring and MAP2-related functional studies were conducted retrospectively as exploratory analyses to generate hypotheses for future validation.

3c. Materials:

(i) Description of the cohort: This paragraph includes considerable detail that is reiterated in the Results section. Condensing this section would improve clarity and avoid redundancy.

Response: We have condensed the cohort description in the Methods (**page 15**) by removing redundant clinical characteristics that are detailed in Table 1 and focusing on essential eligibility criteria and study design elements.

(ii) The following information is missing from the main document

– What types of blood samples were collected?

Response: For both immune subset and CTC analysis, 8 mL peripheral blood samples were collected at baseline, cycle 1 day 15, cycle 3 day 1, and at progression. We have added this information to the Methods (**page 17**) and Extended data file (**pages 3-4**).

– Were the different biopsies obtained from the same lesion in each patient?

Response: We have clarified that on-treatment biopsies were obtained from the same lesion whenever feasible, or from the same organ when re-biopsy of the original lesion was not possible (**page 15**).

3d. Methods:

(i) The section would benefit from improved clarity and completeness. Many abbreviations are introduced without definition, which interrupts readability and makes it difficult to follow the methodological flow. All abbreviations (e.g., CB, NCB) should be written out in full at first mention in both the main text and figure legends.

Response: All abbreviations are now defined at first use in the main text and figure legends.

(ii) Definition of response groups: The categories “clinical benefit (CB),” “no clinical benefit (NCB),” and “exceptional responders” should be explicitly defined in the Materials and Methods section. There is also an inconsistency in the reported number of exceptional responders (three in line 215 versus four in line 188) that should be clarified.

Response: We have added the definitions of the response groups in the Methods section (**page 16**) and clarified the sample numbers: pre-treatment biopsies for genomic and transcriptomic profiling were available for 31 patients in the D+O+C arm (21 CB including 4 exceptional responders, and 10 NCB) and 24 in the D+C arm (14 CB including 3 exceptional responders, and 10 NCB); one additional exceptional responder in the D+C arm was not biopsied due to the absence of measurable lesions on imaging.

(iii) In the paragraph “Single-sample gene set scoring,” essential information about the analyzed samples and the analytical setup (with reference to the Supplementary Material) is

missing. In the current version, it is difficult to understand which data were included in the analysis and how they were selected.

Response: We have revised the Methods section (**pages 16-17**) to specify the number and type of samples analyzed and referenced the Supplementary Materials.

(iv) Multiple analytical approaches were applied to the genomic data. Given that these analyses form the core of the Results section, a more detailed description should be provided here to ensure clarity and context. E.g. Gene set scoring: In the paragraph describing single-sample gene set scoring, please provide details on the genes annotated to the immune and metabolic signatures (20 genes for DOC and 15 for DC, respectively; line 473). These gene lists should also be made available in the main text or shown in Figure 3 to improve transparency.

Response: The gene lists used for the immune and metabolic signatures have been added to the updated Fig. 3. Additional methodological details describing these analyses have also been included in the main Methods section (**pages 16-17**) and Extended data file (**page 3**).

(v) The manuscript should include details of the flow cytometry panels used for immune profiling (antibody list and definition of immune subsets) and for the identification of circulating tumor cells (CTCs), to ensure methodological transparency and reproducibility.

Response: We have added the antibody panels for immune profiling and CTC analyses to a new **Supplementary Table 21**, and the definitions of all immune subsets to a new **Supplementary Table 22**. The Supplementary Methods (**Extended data file, page 4**) now specifies that CTCs were defined as viable, nucleated, EpCAM-positive, CD45-negative cells.

(vi) Immune cell subset analysis: Handling of peripheral blood mononuclear cells (PBMCs) should be described in more detail. Include information on the concentrations of Fc block and live/dead cell markers, and provide a complete antibody list (clone, fluorochrome, and concentration) to ensure reproducibility.

Response: We added details on PBMC handling to the Methods (**page 17**) and provided the full antibody list in a new **Supplementary Table 21**.

(vii) In vitro cell growth and drug sensitivity assays (Figure 7e): Please clarify whether dimethyl sulfoxide (DMSO) was used as a vehicle control and specify its final concentration in the culture medium. The reported growth inhibition appears to be primarily associated with the MAP2 knockout rather than the treatment effect. It would also strengthen the analysis to include PEO1 cells as a positive control.

Response: We have included PEO1 (**new Fig. 7d-f**) and clarified that DMSO (0.01% v/v) together with human isotype IgG (10 µg/mL) was used as the vehicle control (**page 19**).

4. Results

4a. This section should be limited to the presentation of results; interpretation should be reserved for the Discussion.

Response: We have moved interpretative statements to the Discussion where appropriate to maintain a primarily descriptive Results section.

4b. The manuscript would benefit from a more balanced presentation of the data. While the genomic analyses are comprehensively detailed, the phenotypic characterization of peripheral immune cells and circulating tumor cells (CTCs), as well as the functional studies, receive minimal attention and should be expanded for completeness.

Response: We have expanded the Results to ensure a more balanced presentation of our multi-layered data. Specifically, we added detailed descriptions of the systemic reduction in activated and proliferating CD8⁺ and CD4⁺ T-cell subsets and the induction of CD56⁺CD16⁺ NK cells (**page 10**), along with the expanded functional validation of MAP2 (**page 12**).

4c. Currently, some important translational observations are not fully contextualized, and key comparisons are underdeveloped.

(i) IFN- α expression: The reported upregulation of interferon-alpha (IFN- α) in patients with clinical benefit before treatment requires further clarification. The post-treatment increase

of IFN- α observed in both clinical benefit (CB) and no clinical benefit (NCB) groups suggests that this may represent a treatment-related rather than response-specific effect. The authors should reconcile this observation with the relative pretreatment elevation of IFN- α in the CB group (Figure 3a).

Response: We thank the reviewer for highlighting this observation. We have revised the Results (page 9) to clearly distinguish these patterns: post-treatment IFN- α pathway enrichment occurred in both CB and NCB groups, whereas elevated baseline IFN- α signaling was specifically associated with a CB group.

We have expanded the Discussion (page 13) to interpret this finding. We propose that treatment-induced IFN- α upregulation represents a broad pharmacodynamic effect of checkpoint blockade, while baseline IFN- α reflects pre-existing immune priming that may predict response. Only tumors with baseline immune activation possess the microenvironmental features necessary to translate treatment-induced signals into sustained anti-tumor immunity.

(ii) Cytoskeletal and microtubule pathway activation: NCB tumors reportedly show enrichment of cytoskeletal organization and microtubule dynamics pathways (lines 311–312). The biological and morphological relevance of these findings is not talked over (in the Discussion). Were there observable histological or phenotypic differences between these tumors? Since the abstract proposes cytoskeletal profiling as a potential predictive marker for lack of clinical benefit, validation using non-transcriptional methods (e.g., morphological or immunohistochemical assessment) would strengthen this conclusion.

Response: We agree that morphological and immunohistochemical validation would strengthen our conclusions. Unfortunately, residual tissue samples were insufficient for such assessments.

We have expanded the Discussion (page 14) to address the biological relevance of cytoskeletal enrichment, specifically noting that microtubule and cytoskeletal dynamics facilitate structural plasticity and mechanical remodeling that enable tumor cell survival and immune evasion under therapeutic stress⁵⁻⁸.

To provide non-transcriptional validation, we identified microtubule-associated protein 2 (MAP2) as the only gene consistently upregulated in NCB tumors across both our cohort and an independent dataset (GSE206422⁹). Functional studies using siRNA-mediated MAP2 knockdown

in ovarian cancer cell lines with varying MAP2 expression (**new Fig. 7d-f**) demonstrate that MAP2 depletion sensitizes MAP2-high ovarian cancer cells to treatment through both tumor-intrinsic and immune-modulatory mechanisms, providing functional evidence that these cytoskeletal programs drive resistance.

We acknowledge that future studies incorporating spatial transcriptomics and immunohistochemistry will be essential to validate the morphological correlates of these transcriptional signatures and assess MAP2 protein expression in clinical specimens (**page 14**).

(iii) MAP2-related growth effects: The analysis of microtubule-associated protein 2 (MAP2) expression in NCB tumors indicates a role in modulating microtubule dynamics (line 335), yet the rationale for the experimental design is unclear. Given that MAP2 expression is high in OVCAR3 and OVCAR8 cells, why was PEO1 not included as a positive control in the drug-response assays? The observed growth reduction across all knockout models, including untreated and vehicle-treated (dimethyl sulfoxide, DMSO) controls, suggests that the inhibitory effect may derive primarily from the knockout rather than from drug treatment. Clarification is needed to distinguish treatment-specific effects from MAP2-dependent growth suppression.

Response: To distinguish MAP2-dependent baseline effects from treatment-related effects, we performed additional assays in MAP2-high (OVCAR3, OVCAR8) and MAP2-low (PEO1) cell lines. As requested, PEO1 was included as a low-MAP2 control to determine whether the observed phenotypes were baseline-dependent.

We found that MAP2 knockdown reduced baseline proliferation in all three models, with a pronounced effect in MAP2-high cells and a minimal effect in PEO1 (**new Fig. 7e**) indicating the intrinsic growth inhibition is proportional to baseline MAP2 levels. Regarding treatment-specific effects, MAP2 depletion sensitized MAP2-high cells to D+C and D+O+C, yielding additional suppression beyond the baseline knockout effect. In contrast, drug treatment produced no further inhibition in MAP2-depleted PEO1. These results indicate that both intrinsic growth suppression and drug sensitization contribute to the overall phenotype, with the relative contribution of each determined by the cell's baseline MAP2 status. We have revised the Results (**page 12**) to reflect these findings.

5. Figure legends and Figures

5a. Most figure legends require rephrasing for clarity and completeness. Each legend should provide a concise description of the experimental setup, sample size, and main findings to enable understanding without referring to the main text.

Response: Done.

5b. Most of the figures are well designed and effectively illustrate the findings.

(i) Figure 1: The number of analyzed samples (blood and biopsies) should be clearly indicated within the figure itself, not only in the legend.

Response: Fig. 1b now includes the numbers of pre- and on-treatment biopsies and blood samples for each arm directly in the diagram.

(ii) Figure 2: Panels (a) and (b) appear to display identical results using different visualization methods. Please clarify the rationale for presenting both or consider consolidating them.

Response: We retain both panels and now clarify their complementary roles: the waterfall plots depict best tumor response and ORR in RECIST-evaluable patients (Fig. 2a), whereas the swimmer plots depict treatment duration and exceptional responders in the intention-to-treat population (Fig. 2b).

(iii) Figure 3: Panels (g) and (h) are missing—see the figure legend. The Spearman correlation analysis is not convincing (e.g., line 222 $p = 0.35$; line 227 $p = 0.40$), indicating only weak correlations, both for immune and metabolic scores.

Response: We have removed references to panels (g) and (h) from the legend to match the current figure layout. We have also clarified that the correlations between signature scores and PFS are modest and exploratory (**page 8**).

(iv) Figure 4: The observed upregulation appears to reflect a treatment effect rather than a response-associated effect. Please clarify how the post-treatment upregulation of 14- α in both

CB and NCB groups aligns with the relative pretreatment increase of IFN- α in the CB group (Figure 3a).

Response: We have updated the Results (page 9) to indicate that post-treatment IFN- α upregulation appears to be a treatment effect seen in both CB and NCB tumors, whereas higher baseline IFN- α signaling is associated with clinical benefit.

(v) Figure 5: Violin plots are best suited for large datasets. Given the limited sample size, boxplots are recommended for improved clarity and transparency. The number of CB and NCB samples included should be explicitly indicated. Expression markers for m-MDSCs and PMN-MDSCs (panels a and b) should be defined in the figure legend.

Response: We have replaced violin plots with boxplots, added the number of CB and NCB samples to each panel, and defined M-MDSCs (CD11b⁺ CD14⁺ HLA-DR^{low/-} CD15⁻) and PMN-MDSCs (CD14⁻ CD11b⁺ CD15⁺) in the legend.

(vi) Figure 7: The comparison between MAP2A-expressing and knockout (KO) cells across treatment groups would be more intuitive if shown side by side, rather than comparing treatment effects independently. This arrangement would better highlight differences attributable to MAP2A expression status.

Response: New Fig. 7e-f has been reformatted to present MAP2-expressing (siControl) and MAP2-knockdown (siMAP2) cells side by side.

6. Discussion

The Discussion should be more focused and structured. The first paragraph should concisely summarize the main findings. Thereafter, three to four major results should be interpreted individually and in relation to the existing literature before being placed in a broader clinical perspective. Each section should end with a clear point of interpretation rather than a preliminary conclusion. The authors should also integrate the clinical outcomes, immune profiling, and functional data into a more overarching biological and clinical context. As noted in the comments on the Introduction, the recently presented results from the

KEYNOTE-B96 study provide important new clinical data that should be discussed in relation to the present findings. In addition, the translational data require validation, which should be addressed explicitly in this section.

Response: We have restructured the Discussion to improve focus and clarity as suggested.

Reviewer #3

I have focused on evaluation of trial design/methodology, reporting of outcomes, and statistical analyses methods.

1. Abstract

Translational results are highlighted without any data reported - which of these analyses were pre-specified in the protocol/Analysis plan? In the discussion, the exploratory nature of the biomarker analyses is highlighted.

Response: We have revised the Abstract to clarify that transcriptomic profiling was a pre-specified translational endpoint, while additional biomarker analyses were exploratory, aligning with the Discussion.

2. Methods

2a. Study objectives and endpoints: Section states (clinical) endpoints rather than objectives. [ie Primary objective as per clintrials.gov: To determine how effective this combination is in treating ovarian cancer. Then primary endpoint is ORR as measure of antitumour activity]. This section would benefit from specifying pre-specified secondary objectives for translational analyses (and which ones were post-hoc/exploratory).

Response: We have revised the Methods (**page 16**) to distinguish the primary objective from the primary endpoint, specify secondary clinical objectives, and list the pre-specified translational endpoints (WES, RNAseq, immune subset profiling, and CTC analysis). We also now clarify that additional biomarker analyses, including signature scoring and MAP2-related functional studies, were performed as exploratory analyses.

2b. Definition of ORR: complete (CR) or confirmed/unconfirmed PR per investigator-assessed RECIST v1.1. (how was this confirmed? Central review? Second scan?)

Response: We have clarified in the Methods section (**page 16**) that ORR as CR or PR per investigator-assessed RECIST v1.1, including both confirmed and unconfirmed PRs, with confirmation requiring a follow-up scan.

2c. PFS - does it include clinical progression or was it just imaging-defined progression?

Response: We have clarified in the Methods (**page 16**) that PFS included both radiographic and clinical progression, as assessed by the investigators.

2d. In many of the results, and also in the abstract, Clinical benefit (CB: PR or SD>4m) or exceptional responders (PFS>12m) are the key endpoints analysed, but these are not defined in objectives/endpoints (nor are in clintrials.gov). Was this endpoint/objective pre-specified in the trial? Is the duration of SD 'standard' definition for CB?

Response: Thank you for this helpful comment. We have stated that CB (PR or SD \geq 4 months) and exceptional responders (PFS \geq 12 months) were defined retrospectively, explain the rationale (e.g., median PFS and consistency with NRG-GY023¹⁰), and acknowledge that these are post-hoc exploratory endpoints in the Methods (**page 16**).

3. Methods - Statistical analyses

3a. Exact 95%confidence intervals are used for the ORR - but these are not consistent with the 2-stage design. Better inference methods have been advocated: <https://pmc.ncbi.nlm.nih.gov/articles/PMC6047527/>

And are include din R packages, such as clinfun (function twostage.inference).

Similarly, there are methods for proper handling of under-recruitment in phase 2 studies, which allow to 'recalculate' decision thresholds in the 2-stage design: <https://pubmed.ncbi.nlm.nih.gov/25781860/> (I think also implemented in the above R package).

Response: We have recalculated ORR confidence intervals using a design-consistent method for Simon two-stage trials¹¹, and applied a conditional-error approach to account for under-recruitment¹², as suggested. The Methods now describe these procedures and the resulting decision thresholds (page 20).

3b. I am unsure by the attempt to demonstrate statistical equivalence testing between the two treatment arms. This is not an aim/stated in the protocol. I understand the aim of this analysis was to enable the "pooled" analysis of the two cohorts for biomarker analyses. But as designed, it was never aimed at comparing these two treatment groups, patients were not randomised to one or the other, patients had, a priori, different histologies (and selection bias to be allocated into each arm). Was this a pre-specified analysis? Importantly, was the equivalence margin pre-specified a priori? FYI, there are equivalence methods for censored data (but maybe not in the TOSTER package) although I accept with the low %censoring it does not make a big difference. I'd argue that normality test for PFs is 'redundant' as time to event endpoints are usually skewed. In short, I'd remove this whole bit from methods (no linked results) and state all results are presented by cohort as different populations in any case.

Response: We agree that equivalence testing was neither pre-specified nor appropriate for this non-randomized design. All equivalence analyses have been removed, and we present results separately by treatment arm.

3c. For the translational analyses, largely non-parametric tests were used (page 20, lines 584-589). Results indicate that time-to-event analysis were performed (figures 6d and 7b) but this is not explained in the methods. Results clearly show adjustment of multiplicity was implemented (adjusted p-values) but this is not explained in the methods. There is mention to "one-way AMOVA for multiple comparison" in line 591, but I interpreted that this corresponds to the in-vitro studies, is this correct? A reference for the AMOVA method may be useful.

Response: We have added a dedicated paragraph describing the use of Kaplan–Meier and log-rank tests for time-to-event analyses, and we specify that multiple testing for RNAseq analyses

was controlled using the Benjamini–Hochberg method. We also corrected “AMOVA” to “ANOVA” for *in vitro* analyses (page 21).

3d. Lines 591-592 "The $p < 0.05$ were considered significant. All statistical analyses were done using GraphPad Prism v10." Does this refer to the whole analysis or just the in-vitro analysis?

Response: We have clarified that GraphPad Prism v10 was used only for *in vitro* experiments and associated analyses (page 21).

4. Results - Patient enrollment and baseline characteristics

4a. Please report recruitment start/stop dates for each cohort separately.

Response: We have reported it as suggested (page 6).

4b. Results - Efficacy and safety

(i) Please state how many of the PR responses were unconfirmed. When refer to patients having SD, needs to be specified this is 'best overall response as per RECIST - I think only stated in figure 1b legend. Is this something that should be more clear in the endpoint definition?

Response: We appreciate the reviewer’s comment and have revised the Results (page 6). Briefly, in the D+O+C arm, 7 PRs included 1 unconfirmed PR; in the D+C arm, 8 PRs included 2 unconfirmed PRs. We have also clarified in the endpoint definition that SD refers to best overall response per RECIST v1.1.

(ii) This is the main results for cohorts D+C and D+C+O but there is not a statement regarding the primary outcome of the trial - Although it under-recruited, the number of responses observed permit to conclude according to the proposed design: 7PR in 36 D+O+C (planned $\geq 11/37$)  based on ORR, D+O+C did not reach its threshold to investigate

further; 8PR in 27 D+C (planned $\geq 6/35$), so even if recruitment had continued, conclusion would have been that D+C is promising to investigate further (based on ORR).

Response: We thank the reviewer for this helpful clarification. This information has been incorporated into the Results to clearly contextualize the clinical activity of each arm (page 6).

5. Results - Upregulation of immune and metabolic pathways

5a. I would be explicit in stating "In both treatment arms, tumors from CB group (D+O+C [n = 21], D+C [n = 14]) exhibited significant upregulation of interferon alpha response than tumours in NCB group (all adjusted p...)"

Response: We have revised the text as suggested (page 7).

5b. [Editorial note: please note that the reviewer has confidentially highlighted how it would be important to clarify whether the translational analyses - and the signals identified in the paper - are influenced by the patients being platinum sensitive vs platinum refractory. Please also ensure to address/discuss this point].

Response: We have expanded our analysis to ensure that the identified signals are not confounded by prior platinum sensitivity. By restricting our analysis to the platinum-resistant subpopulation, we demonstrate that our signatures maintain high discriminatory power with the Area Under the Receiver Operating Characteristic curve (AUC) values > 0.7 interpreted as strong and > 0.8 as excellent predictive accuracy (new Supplementary Table 20 and Extended Data Fig. 3a-c):

- Immune signature maintained strong predictive accuracy in identifying CB within the platinum-resistant D+O+C arm (AUC = 0.756, $p = 0.031$) and the external GSE206422⁹ dataset (AUC = 0.81, $p = 0.033$).
- NCB signature showed excellent performance in predicting NCB within platinum-resistant subgroups of D+C (AUC = 0.90, $p = 0.005$) and D+O+C arms (AUC = 0.806, $p = 0.009$), with a consistent trend observed in the GSE206422 dataset (AUC = 0.714, $p = 0.207$).
- Metabolic signature specifically preserved excellent predictive value for identifying CB in the platinum-resistant D+C subgroup (AUC = 0.843, $p = 0.019$).

These results demonstrate that the molecular features captured by our translational signatures represent intrinsic therapeutic vulnerabilities unlikely driven by platinum sensitivity. We have updated the Results (**page 11**) to highlight these findings.

Reviewer #4

Response: Thank you.

References

- 1 Wallin, J. J. *et al.* Atezolizumab in combination with bevacizumab enhances antigen-specific T-cell migration in metastatic renal cell carcinoma. *Nat Commun* **7**, 12624 (2016). <https://doi.org/10.1038/ncomms12624>
- 2 Simula, L. *et al.* Mitochondrial metabolism sustains CD8(+) T cell migration for an efficient infiltration into solid tumors. *Nat Commun* **15**, 2203 (2024). <https://doi.org/10.1038/s41467-024-46377-7>
- 3 Hu, T. *et al.* Metabolic regulation of the immune system in health and diseases: mechanisms and interventions. *Signal Transduct Target Ther* **9**, 268 (2024). <https://doi.org/10.1038/s41392-024-01954-6>
- 4 Colombo, N. *et al.* LBA3 Pembrolizumab vs placebo plus weekly paclitaxel ± bevacizumab in platinum-resistant recurrent ovarian cancer: Results from the randomized double-blind phase III ENGOT-ov65/KEYNOTE-B96 study. *Annals of Oncology* **36**, S1697 (2025). <https://doi.org/10.1016/j.annonc.2025.09.049>
- 5 Carmeliet, P. & Jain, R. K. Molecular mechanisms and clinical applications of angiogenesis. *Nature* **473**, 298-307 (2011). <https://doi.org/10.1038/nature10144>
- 6 Yetkin-Arik, B. *et al.* Angiogenesis in gynecological cancers and the options for anti-angiogenesis therapy. *Biochim Biophys Acta Rev Cancer* **1875**, 188446 (2021). <https://doi.org/10.1016/j.bbcan.2020.188446>

- 7 Horikawa, N. *et al.* Anti-VEGF therapy resistance in ovarian cancer is caused by GM-CSF-induced myeloid-derived suppressor cell recruitment. *Br J Cancer* **122**, 778-788 (2020). <https://doi.org/10.1038/s41416-019-0725-x>
- 8 Friedl, P. & Alexander, S. Cancer invasion and the microenvironment: plasticity and reciprocity. *Cell* **147**, 992-1009 (2011). <https://doi.org/10.1016/j.cell.2011.11.016>
- 9 Rosario, S. R. *et al.* Integrative multi-omics analysis uncovers tumor-immune-gut axis influencing immunotherapy outcomes in ovarian cancer. *Nat Commun* **15**, 10609 (2024). <https://doi.org/10.1038/s41467-024-54565-8>
- 10 Lee, J. M. *et al.* Comparing Durvalumab, Olaparib, and Cediranib Monotherapy, Combination Therapy, or Chemotherapy in Patients with Platinum-Resistant Ovarian Cancer with Prior Bevacizumab: The Phase II NRG-GY023 Trial. *Clin Cancer Res* **31**, 2370-2378 (2025). <https://doi.org/10.1158/1078-0432.CCR-24-3877>
- 11 Koyama, T. & Chen, H. Proper inference from Simon's two-stage designs. *Stat Med* **27**, 3145-3154 (2008). <https://doi.org/10.1002/sim.3123>
- 12 Englert, S. & Kieser, M. Methods for proper handling of overrunning and underrunning in phase II designs for oncology trials. *Stat Med* **34**, 2128-2137 (2015). <https://doi.org/10.1002/sim.6479>

Reviewer #1

I'm satisfied with author responses to my comments. I have no further comments.

Response: Thank you.

Reviewer #2

NCOMMS-25-76416A

The manuscript has improved substantially following revision. The majority of our previous comments have been addressed, and the necessary modifications have been implemented. Overall, the revisions have improved the clarity, coherence, and scientific presentation of the study.

Major comment:

It remains unclear why the authors chose to compare only the D+O+C arm with the D+C arm, while excluding the D+O arm—which, based on the rationale presented in the Introduction, would appear to be a more logical comparator—or why all three treatment arms were not included in the comparative analyses. This issue requires clearer justification and should be addressed consistently throughout the different sections.

Response: We thank the reviewer for this comment and appreciate the opportunity to clarify the study design and analytical framework.

This study was designed as a multi-treatment arm pilot study (like an umbrella study) with a Simon's two-stage design for signal-finding, rather than a randomized study of comparing 3 different treatment arms with prespecified statistical endpoints. As a result, each arm enrolled distinct patient populations, making direct comparative analyses methodologically inappropriate. Therefore, all analyses were conducted in an arm-specific, non-comparative manner. We clarified the current study as a non-randomized study in **pages 3 and 5**.

We also evaluated the technical feasibility of joint translational analyses and consulted with the institutional sequencing core facility. To avoid batch effects, DNA and RNA samples from the D+C and D+O+C arms were prepared using the same methods and within the same time frame, and whole-exome sequencing (WES) and RNA sequencing were processed in parallel using the same platforms (NovaSeq Xplus), library preparation kits (exome V8 and RNA Ribo-Zero Plus) and workflows. Immune subset

profiling and circulating tumor cell (CTC) analyses for these two arms were also performed in parallel using the same antibody lots.

In contrast, DNA/RNA sample preparation, WES, RNA sequencing, and immune/CTC analyses for the D+O arm¹ were performed before COVID19 (2018-2019), using different sequencing platform (NovaSeq 6000), library preparation kits (exome V7 and RNA TruSeq), target capture designs, data processing pipelines (including software versions and reference annotations), and antibody lots. Although Illumina sequencing platforms themselves are not expected to introduce major quality-related batch effects, the use of non-overlapping workflows, software, and reagents, together with the distinct processing periods (2018-2019 vs. 2025), would introduce compound batch effects that cannot be reliably separated from true biological signals.

For these reasons, inclusion of the D+O arm in joint or comparative translational analyses would be methodologically inappropriate. Accordingly, all analyses were performed in an arm-specific, non-comparative manner, as clarified in the Methods (**page 16**).

Minor comments:

1. Abstract

1a. The cohorts should be presented in the same order throughout the paragraph.

Response: The cohorts have been reordered throughout the Abstract to ensure consistent presentation (D+O+C followed by D+C).

1b. The last sentence should be rephrased. Suggested revision: “These findings support proof-of-concept clinical activity of D+O+C and D+C and identify molecular signatures with potential predictive value in subsets of recurrent EOC.”

Response: We have revised it as suggested.

2. Introduction OK

Response: Thank you.

3. Methods

3a. Please rephrase the description of the translational research end-points to better reflect the results from analyses performed and the results obtained (page 16, lines 454–457).

Response: The Methods section has been revised to better align with the analyses performed and results presented (**page 16**).

3b. Given that only 13 genes are annotated to the metabolic signature (Supplementary Table 11 and Figure 3e), the current description in the Methods section requires revision

Response: It has been corrected.

3c. Immune cell subset analysis: When describing the handling of peripheral blood mononuclear cells (PBMCs), the concentration of the Fc-blocking agent could be reported rather than the volume used (page 17)

Response: The Fc receptor blocking reagent used (Miltenyi Biotec, #130-059-901) is supplied without a specified concentration. As recommended by the manufacturer, the reagent was used at 20 μL per 1×10^7 cells (or fewer). We have clarified this in the Methods section (**page 17**).

4. Results

4a. Main document: The Figures have been included twice.

Response: The Figures are now uploaded separately in the submission system to avoid duplication.

4b. The following statement “The concomitant upregulation of angiogenesis and ECM genes, alongside downregulation of chromatin and endothelial transcripts, suggests a state of immune exclusion with enhanced tissue plasticity” (page 10) could be removed and eventually moved to the Discussion.

Response: This statement has been removed from the Results section, and its conceptual implications are addressed in the Discussion (paragraph 3, **page 13**).

5. Figure legends and Figures

5a. Figure 6a: Add “NCB” to improve clarity.

Response: Done.

5b. Significant IFN- α upregulation: Include the p-values to the text (page 7).

Response: The specific adjusted p-values have been added (**page 7**).

5c. Figure 5: The p-values for the NCB group in the M-MDSC analyses should be reported in the text, as a trend toward M-MDSC downregulation at C1D15 is also observed in this group.

In addition, Figure 5f is not described in the Results section; if this panel is redundant, it should be removed from Figure 5 (for example, replaced with the CTC results shown in Extended Figure 2a/b).

Response: We have reported the p-values for M-MDSC changes in the NCB groups in the Results (**page 10**), showing a non-significant downward trend ($p = 0.1055$ in D+O+C and $p = 0.2656$ in D+C).

Fig. 5f has been removed to avoid redundancy, and the CTC analyses are retained in Extended Data Fig. 2.

6. Discussion

The relevance of the following statement is unclear: “Our data showing that MAP2 knockdown sensitizes cells to treatment may be clinically relevant given the KEYNOTE-B96/ENGOT-ov65 trial4 (pembrolizumab, bevacizumab plus paclitaxel) reported improved PFS and OS in patients with recurrent EOC, although this requires further mechanistic studies and prospective clinical validation.” The mechanistic link between the presented preclinical findings and the clinical outcomes reported in this trial is not sufficiently explained and requires clearer justification.

Response: We agree and have revised the Discussion to remove speculative statements regarding clinical relevance (**page 14**). The trial is now cited as a hypothesis-generating clinical context, and the clinical relevance of the MAP2 findings requires further mechanistic validation.

7. Others

Clinical benefit (CB) and no clinical benefit (NCB) should be defined at first mention.

Response: Done.

Reviewer #3

Thank you - the authors have addressed my comments and concerns.

Response: Thank you.

Reviewer #4

Response: Thank you.

References

- 1 Lampert, E. J. *et al.* Combination of PARP Inhibitor Olaparib, and PD-L1 Inhibitor Durvalumab, in Recurrent Ovarian Cancer: a Proof-of-Concept Phase II Study. *Clin Cancer Res* **26**, 4268-4279 (2020). <https://doi.org/10.1158/1078-0432.CCR-20-0056>

Reviewer #2

The authors have addressed all comments and concerns.

Response: Thank you.

Reviewer #4

Response: Thank you.